# Faster Query Times for Fully Dynamic k-Center Clustering with Outliers

**Leyla Biabani**
Eindhoven University of Technology
Eindhoven, The Netherlands
`l.biabani@tue.nl`

**Annika Hennes**
Heinrich Heine University Düsseldorf
Düsseldorf, Germany
`annika.hennes@hhu.de`

**Morteza Monemizadeh**
Eindhoven University of Technology
Eindhoven, The Netherlands
`m.monemizadeh@tue.nl`

**Melanie Schmidt**
Heinrich Heine University Düsseldorf
Düsseldorf, Germany
`mschmidt@hhu.de`

## Abstract

Given a point set $P \subseteq M$ from a metric space $(M, d)$ and numbers $k, z \in \mathbb{N}$, the *metric $k$-center problem with $z$ outliers* is to find a set $C^* \subseteq P$ of $k$ points such that the maximum distance of all but at most $z$ outlier points of $P$ to their nearest center in $C^*$ is minimized. We consider this problem in the fully dynamic model, i.e., under insertions and deletions of points, for the case that the metric space has a bounded doubling dimension $\dim$. We utilize a hierarchical data structure to maintain the points and their neighborhoods, which enables us to efficiently find the clusters. In particular, our data structure can be queried at any time to generate a $(3 + \varepsilon)$-approximate solution for input values of $k$ and $z$ in worst-case query time $\varepsilon^{-O(\dim)} k \log n \log \log \Delta$, where $\Delta$ is the ratio between the maximum and minimum distance between two points in $P$. Moreover, it allows insertion/deletion of a point in worst-case update time $\varepsilon^{-O(\dim)} \log n \log \Delta$. Our result achieves a significantly faster query time with respect to $k$ and $z$ than the current state-of-the-art by Pellizzoni, Pietracaprina, and Pucci [18], which uses $\varepsilon^{-O(\dim)}(k+z)^2 \log \Delta$ query time to obtain a $(3 + \varepsilon)$-approximate solution.

## 1 Introduction

Clustering is a fundamental problem in machine learning and it has applications in many areas ranging from natural to social sciences. As a basic unsupervised learning method, it allows us to find structure in data. Classical center-based clustering methods for objectives like $k$-means, $k$-median and $k$-center have been around for many decades [9, 13, 16]. Since the beginning of clustering, the data to be analyzed has changed dramatically, raising new challenges for clustering methods. Data nowadays comes in large to huge batches and is often prone to inherent change: Content on social media platforms is constantly added, re-organized or deleted, streaming services handle an always ongoing flow of people starting and stopping to watch, tweets are the fastest method to distribute news – and delete them later if desired. Data analysis methods thus face the challenge of analyzing huge amounts of data, ideally permanently updating their solution.

In this paper, we consider the problem of *dynamic $k$-center clustering *with outliers*. The (metric) $k$-center problem is to find a set $C^* \subseteq P$ of $k$ points such that the maximum distance of all points of $P$ to their nearest center in $C^*$ is minimized. This objective is well understood, 2-approximations for the problem are known [9, 13] and better approximations are not possible unless $P$ equals $NP$ [14].

37th Conference on Neural Information Processing Systems (NeurIPS 2023).

Table 1: Fully dynamic $k$-center algorithms *without outliers*.

| Year | Ref. | Metric | Update time | Approx |
|------|------|--------|-------------|--------|
| 2018 | [3] | general | amort. $O(k^2 \frac{\log \delta}{\varepsilon})$ | $2 + \varepsilon$ |
| 2019 | [19] | Eucl. $\mathbb{R}^d$ | amort. $O(\varepsilon^d \log \Delta \log n)$ | 16 |
| 2021 | [10] | doubl. | $\varepsilon^{-O(\dim)} \log \Delta \log \log \Delta$ | $2 + \varepsilon$ |
| 2023 | [1] | general | amort. $O(k \operatorname{polylog}(n, \Delta))$ | $2 + \varepsilon$ |

Considering outliers for $k$-center is very natural. Since $k$-center minimizes the maximum radius, a single measuring error can destroy the structure completely. Thus, the $(k, z)$-center problem allows us to ignore $z$ points. These belong to no clusters but are deemed to be *outliers*, and so they are ignored when determining the maximum radius of a cluster. The concept was introduced in 2001 by Charikar, Khuller, Mount, and Narasimhan [6] who as one result provide a 3-approximation for $k$-center with outliers. This variant has since been the subject of many papers, and the best-known approximation algorithm for it, which gives a 2-approximation, was derived as recently as 2016 [2].

In parallel, fully dynamic $k$-center algorithms (without outliers) have been developed [1, 3, 8, 10] which work with a changing point set given as a stream of insertions and deletions. All these maintain a data structure that allows for continual updates to keep adapting to the changing data. An overview is given in Table 1. All results in this table support some form of *membership queries*: Given a point, return the cluster / center of the cluster of that point in the approximate solution. This type of query can typically be answered in time $O(1)$. Reporting the centers takes more time or is not necessarily explained how to do.

Now the challenge is to combine the two parallel developments into a fully dynamic algorithm for $(k, z)$-center clustering. We discuss the setting more precisely and then our results and related work.

**Our setting** We say that a metric space has *doubling dimension* $\dim$ if $\dim$ is the smallest positive integer such that any ball of radius $r$ can be covered by at most $2^{\dim}$ balls of radius $r/2$. For an overview of this concept, we also refer the reader to [12, 11]. Let $(M, d)$ denote such a metric space and let $P \subseteq M$ be a set of points. By $\mathrm{d_{min}} := \inf\{d(x, y) \mid x, y \in P, x \neq y\}$ and $\mathrm{d_{max}} := \sup\{d(x, y) \mid x, y \in P\}$ we denote the minimum and maximum inter-point distance within $P$, respectively. Without loss of generality, we assume that $\mathrm{d_{min}} = 2$. Let $\Delta := \mathrm{d_{max}} / \mathrm{d_{min}}$ be the aspect ratio of $P$.

We are observing a dynamic stream, i.e., we start with a point set $P = \emptyset$ and then process a sequence of operations whose length is unknown a priori. Each operation either adds a point from $M$ to $P$ or it deletes a point which is currently in $P$ from $P$. We assume only deletions of points that are currently present are allowed, and we also assume that we can always compute $d$ in time $O(1)$ (otherwise, multiply all times mentioned in this paper by the time necessary to make one distance computation). We refer to the point set after time $t$ (i.e., after $t$ operations happened) as $P_t$. If it is clear from the context, we will sometimes drop the subscript. We assume that $d_{\max}$ and $d_{\min}$ are fixed throughout the whole sequence of updates, i.e. $d_{\max} = \sup\{d(x, y) \mid x, y \in P_t, t \geq 0\}$, and analogously for $d_{\min}$. Therefore, $\Delta$ is an upper bound on the aspect ratio over all updates and is known in advance. This is in accordance with a sequence of other works [1, 4, 3, 7, 10, 19].

The problem that we address is the dynamic *$k$-center problem with $z$ outliers* or in short $(k, z)$-*center problem*: Given $P_t$ and numbers $k$ and $z$, produce a set $C \subseteq P_t$ of size at most $k$ such that the maximum distance of all but at most $z$ points to their nearest neighbor is minimized. More formally, $\min_{Z \subseteq P_t, |Z| \leq z} \max_{x \in P_t \setminus Z} \min_{c \in C} d(x, c)$ is minimized among all possible choices of $C$ with $|C| \leq k$. [1]

---

[1] Notice the small detail that we do not ask to compute $Z$ but only to provide centers that are good centers under the assumption that at most $z$ points can be ignored. In the offline setting, finding $Z$ can be done by assigning every point in $P$ to its closest point in $C$ and then ignoring the $z$ points farthest away. In the dynamic setting and with our data structure, we could also produce the outliers. However, we believe that the outliers are less interesting than the centers, and since there could be substantially many outliers, we rather only report the cluster centers to avoid a linear dependence on $z$.

Table 2: Fully dynamic $k$-center algorithms *with outliers* and their update/query times.

| Year | Ref. | Metric | Update time | Query time | Approx |
|------|------|--------|-------------|------------|--------|
| 2022 | [4] | general | $O(\frac{1}{\varepsilon}(k+z)^2 \log \Delta)$ | $O(\frac{1}{\varepsilon}(k+z)^2 \log \Delta)$ | $14+\varepsilon$ |
| 2023 | [18] | doubl. | $\frac{1}{\varepsilon^{O(\text{dim})}} \log \Delta$ | $\frac{1}{\varepsilon^{O(\text{dim})}}(k+z)^2 \log \Delta$ | $3+\varepsilon$ |
| 2023 | [7] | doubl. | $O(\frac{k}{\varepsilon^{\text{dim}}} + z) \log^4(\frac{k\Delta}{\varepsilon\delta})$ | $O((\frac{k}{\varepsilon^{\text{dim}}} + z)^2 k \log(\frac{k}{\varepsilon^{\text{dim}}} + z))$ | $3+\varepsilon$ |
| now | Thm 1.1 | doubl. | $\frac{1}{\varepsilon^{O(\text{dim})}} \log n \log \Delta$ | $\frac{1}{\varepsilon^{O(\text{dim})}} k \log n \log \log \Delta$ | $3+\varepsilon$ |

Our focus is on building a dynamic algorithm that has a low update time for insertion and deletions. We are allowed to store $P$ (and additional information). Our aim is to achieve an update time for our data structure that is independent of $k$ and a query time that is at most linear in $k$. For this, it is necessary to consider a restricted model, as Bateni *et al.* [1] showed that in arbitrary metric spaces, an update time of $\Omega(k)$ is necessary. We consider metric spaces *of bounded doubling dimension*. Obviously, time $\Theta(k)$ is necessary to return an actual solution, so we distinguish between *update time* for insertion and deletions of points and *query time* for obtaining a solution from the data structure.

**Our result compared to related work**     We present a deterministic $(3+\varepsilon)$-approximation algorithm for the $k$-center problem with $z$ outliers in bounded doubling dimension that is fully dynamic, i.e., points can be inserted and deleted. Our algorithm is based on a data structure that requires space linear in $n$ and does not need to know $k$ and $z$ in advance. As a result, we can answer queries for all $k$ and $z$, or in other words, $k$ and $z$ can be part of the query. Moreover, our algorithm does not require knowledge of the doubling dimension $\text{dim}$ of the underlying metric space, and it only appears in the analysis.

**Theorem 1.1** (Main theorem).  *Let $(M, d)$ be a metric space of bounded doubling dimension $\text{dim}$, and let $\varepsilon > 0$ be an error parameter. There exists a deterministic dynamic algorithm that allows the insertion or deletion of points from $M$ using worst-case $\varepsilon^{-O(\text{dim})} \log n \log \Delta$ update time. Moreover, at any time $t$, it can be queried by parameters $k$ and $z$ to compute a $(3+\varepsilon)$-approximate solution for the $k$-center problem with $z$ outliers of $P_t$ using worst-case $\varepsilon^{-O(\text{dim})} k \log n \log \log \Delta$ query time, where $P_t$ is the set of points that are inserted but not deleted up to time $t$, and $n$ is the size of $P_t$.*

The first approximation algorithm for this problem has been obtained by Chan, Lattanzi, Sozio, and Wang in 2022 [4], which has an approximation ratio of $14 + \varepsilon$. Very recently and in independent work, Pellizzoni, Pietracaprina, and Pucci [18] derived an algorithm with approximation ratio $3 + \varepsilon$. For the problem of distributed $k$-center with outliers, De Berg, Biabani, and Monemizadeh [7] give a randomized dynamic algorithm that also achieves a guarantee of $3 + \varepsilon$ with failure probability $\delta$. Both of these and also we develop algorithms that answer *solution queries* rather than membership queries, i.e., they can produce a center set at any time and this set is an approximately good solution for the current $k$-center with outliers instance. We state the update times (for insertions and deletions of points) and the query times (for reporting the current approximate solution) of these and our approaches in Table 2.

All three known results have a dependency on $(k + z)^2$ in their query time which we improve to a linear dependence on $k$ and no dependence on $z$. This gives our approach an advantage even if $k$ or $z$ are only mildly dependent on $n$: Already for $k \in O(\log n)$ and $z \in O(\sqrt{n})$, the query complexity of [18] would be linear in $n$ compared to the poly-logarithmic dependency in terms of $n$ for our query time. More interestingly, for a realistic regime where $z$ is an epsilon-fraction of $n$, the query time of the dynamic algorithm presented in [18] has quadratic dependency in terms of $n$ while our query time still has a logarithmic dependency in terms of $n$ and linear dependency in terms of $k$. [4] also states a bicriteria approximation where the number of outliers can be violated with improved running time, but the dependency on $k$ is still quadratic. When comparing to [7], it should be noted that their algorithm is only applicable to the Euclidean metric space, is randomized and only works against oblivious adversaries. Further, the coreset size (and the space complexity) of the algorithm in [7] is $x = O(k\epsilon^{-\text{dim}} \log \Delta + z)$. In the streaming model, in order to extract the solution, one needs to run an offline algorithm on the coreset. In this way, the query time of this algorithm is $O(kx^2 \log x) = O(k(z^2 + k^2)\epsilon^{-2dim} \log^2 \Delta)$ which is significantly worse than the query time of our algorithm.

The main difference of our approach is that it makes the greedy algorithm for $k$-center with outliers dynamic while the previous dynamic algorithms extract a coreset and run the greedy algorithm on this coreset. Known dynamic algorithms [4, 7, 18] for the $k$-center problem with outliers maintain a coreset after every update. In particular, in [18], they extract the coreset by simply reading the solution from the cover tree. To extract an approximate solution for this problem, one needs to run a known (offline) greedy algorithm on this coreset. In this way, the query complexities of those algorithms are dominated by the running time of the greedy algorithm. The novelty of our dynamic algorithm is that we make the greedy algorithm itself dynamic. To this end, we use heap data structures to compute a ball that covers the maximum number of points, and dynamic neighborhood sets to obtain points in the expanded maximum ball and update their corresponding keys in the heap to recursively find the next maximum balls.

**Navigating nets.** The state-of-the-art algorithm for the fully dynamic $k$-center problem (without outliers) in doubling metrics, developed by Goranci et al. [10] uses *navigating nets* originally introduced by Krauthgamer and Lee [15]. The basic idea of a navigating net is to start with the point set $P_t$ as the base net $N_1$ and then compute coarser and coarser variants of it, resulting in a hierarchy of nets $N_1, N_2, \ldots, N_{2^i}, \ldots$. For every level in this hierarchy, the points are good representatives of the points in the level below but the higher we get in the net, we have fewer points that are better separated. More precisely, following Krauthgamer and Lee [15], every level is a so-called *$r$-net* of the level below it. An $r$-net on a set $X$ is a subset $Y \subseteq X$ that satisfies two properties:

1. $\forall p \in X \exists y \in Y : d(p, y) < r$ (Covering)
2. $\forall x, y \in Y : d(x, y) \geq r$ (Packing)

In a navigating net, one always assures that $N_r$ is an $r$-net of $N_{r/2}$. Assume that the minimum inter-point distance $d_{min}$ satisfies $d_{min} = 2$ and that the ratio between maximum and minimum inter-point distance is $\Delta$. Then a navigating net with $O(\log \Delta)$ levels can be computed in a greedy fashion. Intriguingly, one can show that in the resulting navigating net, the lowest level with $|N_r| \leq k$ provides an 8-approximation for the $k$-center problem (without outliers, i.e., $z = 0$). This was first observed (in the context of streaming algorithms and there known as the *doubling algorithm*) by Charikar, Chekuri, Feder, and Motwani [5]. An additional trick to improve the approximation ratio was discovered by McCutchen and Khuller [17] (again, in the setting of streaming). They observed that the quality of the solutions depends heavily on what the radius of the lowest level is, and that this radius can be shifted by small amounts, and one of the shifted versions will give a $2 + \varepsilon$ approximation. By maintaining the navigating net structure and using a similar shifting trick, dynamic $k$-center algorithms maintain enough information to answer queries by checking one or a few levels of the resulting navigating net and reporting the points on that level as the solution, as for example in [1, 10].

**Challenges with outliers.** For the $(k, z)$-center problem, the challenge is that there is no level in the navigating net which provides a good solution in itself. We know that an optimal $(k, z)$-center solution has a radius that lies between the radius of a $k$-center solution (because any $k$-center solution is feasible for the $(k, z)$-center problem) and the radius of a $(k + z)$-center solution (because any $(k, z)$-center solution is feasible for the $(k + z)$-center problem). But as Figure 1 shows, neither of the two choices may provide a good estimate for the optimum $(k, z)$-center solution. Say level $r_1$ is the first to satisfy $|N_{r_1}| \leq k + z$ and level $r_2$ is the first to satisfy $|N_{r_2}| \leq k$. Then the points in $N_{r_2}$ are a feasible solution for the $(k, z)$-center problem, but their radius may be too large, while the radius of the solution in $N_{r_1}$ is guaranteed to be small enough, but these points may not constitute a feasible solution for $(k, z)$-center. This is because every point in $N_r$ represents several points in the levels below it. So we cannot simply divide the $k + z$

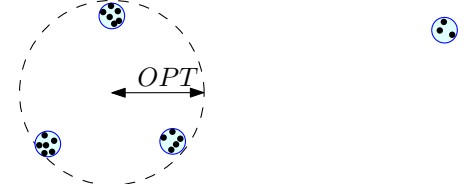

Figure 1: For $k = 1$ and $z = 3$, $k$-center and $(k + z)$-center provide bad estimations for the best $(k, z)$-center solution in this example. We see: (a) Dashed Ball: Optimum solution for $(k, z)$-center. (b) Small Balls (too small): Optimum solution for $(k+z)$-center. (c) Not drawn: the optimum solution for $k$-center is a ball around all points (too large).

points into $k$ centers and $z$ outliers (see Figure 1). Therefore, there is no individual level which we can query to obtain a solution. Instead, we need to compute a solution from the navigating net levels while making sure not to lose too much in terms of quality and query time.

Pellizzoni *et al.* [18] base their approach on the so-called "cover tree" data structure, which is similar to that of a navigating net, to compute a coreset from which they construct the final solution. In their approach, they find the lowest level with $|N_r| \leq 2^{O(\dim)} \varepsilon^{-\dim}(k+z)$ and demonstrate that such $N_r$ serves as an $\varepsilon$-coreset for $(k, z)$-center. However, since it is not a feasible solution, they utilize a 3-approximation greedy algorithm to obtain a $(3 + \varepsilon)$-approximate clustering, which requires a time complexity of $\Omega(\varepsilon^{-2\dim}(k+z)^2)$. In this paper, we strategically integrate additional information at each level to directly emulate the offline 3-approximation greedy algorithm by Charikar *et al.* [6] in the dynamic setting. This approach enables us to achieve a time complexity that is linearly dependent on $k$ and remarkably independent of $z$.

## 2 Our data structure

In this section, we describe our data structure, which has $O(\log \Delta)$ levels. In each level $r$, we have net $N_r$, which is an $r$-net of $N_{r/2}$. For each point $p \in N_r$, we keep its weight $w_r(p)$ and local neighborhoods $B_{p,r}^1$ and $B_{p,r}^3$, which are balls of radius $(1 + O(\varepsilon))r$ and $(3 + O(\varepsilon))r$ respectively. Then we maintain max-heap $H_r$ which is designed to store the total weights of neighborhoods $B_{p,r}^1$ for all $p \in N_r$. We also keep $\text{rep}_r(p)$ for each point $p \in P$, which is the representative of $p$ in $N_r$. Below, we provide a detailed explanation for elements of our data structure.

**Input and set $R$.** To start building our data structure, we require three input parameters: $\Delta$, $\varepsilon'$, and $\beta$, where $0 < \varepsilon' < 1$, and $\frac{1}{2} < \beta \leq 1$. $\Delta$ denotes the spread ratio of the underlying point set $P$ and $\varepsilon'$ is the given error parameter. We utilize $\beta$ to define $R := \{0\} \cup \{\beta \cdot 2^\ell \mid 0 \leq \ell \leq \lceil \log_2 \Delta \rceil + 1, \ell \in \mathbb{Z}\}$ as a set of levels. Additionally, we define $\varepsilon := 2^{\lfloor \log \varepsilon' \rfloor}$. This implies that $\varepsilon'/2 < \varepsilon \leq \varepsilon'$. Furthermore, if $r \in R$, then either $\varepsilon r \in R$ or $\varepsilon r < \beta$.

**Nets $N_r$.** We maintain net $N_r$ for each $r \in R$, satisfying the following conditions: $N_0 = N_\beta = P_t$, where $P_t$ is the set of points at time $t$, and for $r > \beta$, $N_r$ is an $r$-net of $N_{r/2}$. For $p \in N_r$ and $q \in N_{r/2}$, we say $p$ is a *parent of* $q$ (and $q$ is a *child of* $p$) if $d(p, q) \leq r$. In our data structure, we store all the parents and children for each point for all the nets. It is important to note that each point can have at most $2^{O(\dim)}$ parents or children. In the full version, we explain how to handle each insert/delete in time $2^{O(\dim)} \log \Delta$.

During the paper, we may refer to $N_{\varepsilon r}$ for $0 < \varepsilon r < \beta$, which means $\varepsilon r \notin R$. Besides, note that since $\text{d}_{\min} = 2$ and $\beta < 2$, $P_t$ is an $r$-net for any $0 \leq r \leq \beta$. To solve this issue, we define $N_r := P_t$ for any $0 < r < \beta$, however, we do not explicitly keep any $r$-nets for $0 < r < \beta$ in our data structure to prevent redundancy.

**Representatives $\text{rep}_r$ and weights $w_r$.** For all $r \in R$, we inductively define functions $\text{rep}_r \colon P_t \to N_r$ such that each point $p \in P_t$ has a unique *representative* $\text{rep}_r(p)$ in net $N_r$. To define $\text{rep}_r(p)$, we consider three cases. First, if $p \in N_r$, then we define $\text{rep}_r(p) := p$. Second, if $p \notin N_r$ but $p \in N_{r/2}$, we uniquely choose one of the parents of $p$ in $N_r$ as $\text{rep}_r(p)$. Third, if $p \notin N_{r/2}$ and therefore $p \notin N_r$, we define $\text{rep}_r(p) = \text{rep}_r(\text{rep}_{r/2}(p))$. Again to avoid inconsistency, we define $\text{rep}_r(p) := p$ for any $0 < r < \beta$, but we do not store this explicitly. We show in Lemma 2.1 that the distance between each point $p$ and its representative in $N_r$ is at most $2r$.

Next, we define a weight function $w_r$ for every level. For every point $p \in N_r$, its *weight* $w_r(p)$ denotes the number of points that $p$ is representing in $N_r$. More formally, $w_r(p) = \left| \{ q \in P_t \mid \text{rep}_r(q) = p \} \right|$. For a set $X \subseteq N_r$, we define $w_r(X) := \sum_{p \in X} w_r(p)$. Equivalently, $w_r(X) = \left| \{ y \in P_t \mid \text{rep}_r(y) \in X \} \right|$. Note that the sum of weights of all representatives at any level is equal to the total number of points, so $w_r(N_r) = |P_t|$ for all $r \in R$.

Since we store all the parents and children, we can easily update the representatives and therefore the weights after each insert/delete. The details are presented in the full version.

**Balls $B^1$ and $B^3$.**    Let $r \in R$. For every point $p \in N_{\varepsilon r}$, we maintain the *close neighborhood* $B^1_{p,r} := \mathrm{Ball}(p, (1+4\varepsilon)r) \cap N_{\varepsilon r}$ and the *extended neighborhood* $B^3_{p,r} := \mathrm{Ball}(p, (3+8\varepsilon)r) \cap N_{\varepsilon r}$, where $\mathrm{Ball}(p, \lambda) = \{q \in M \mid d(p,q) \le \lambda\}$ denotes the ball of radius $\lambda$ around $p$ in the metric space $M$. Then $B^1 := \{B^1_{p,r} \mid r \in R, \; p \in N_{\varepsilon r}\}$ and $B^3 := \{B^3_{p,r} \mid r \in R, \; p \in N_{\varepsilon r}\}$ are the collections of local neighborhoods at different scales. As mentioned in Lemma 2.2, the size of each of these neighborhoods is at most $2^{O(\mathrm{dim})}/\varepsilon^{\mathrm{dim}}$. We also show that we can update these neighborhoods in time $2^{O(\mathrm{dim})}\varepsilon^{-2\,\mathrm{dim}} \log \Delta$ after each insert/delete in the proof of Lemma 2.3.

**Heaps $H_r$.**    For every $r \in R$, we maintain a max-heap $H_r$ of the set $\{(p, w_{\varepsilon r}(B^1_{p,r})) \mid p \in N_{\varepsilon r}\}$ representing the weight of the close neighborhood of any point in level $r$. With $H_r[p]$ we denote the value of the max heap element with key $p$. Since $B^1_{p,r}$ might change, we need to find the position of the respective heap element by its key $p$. Hence, for any level $r$, we keep a pointer from every element in $N_r$ to its corresponding element in $H_r$.

Whenever an update happens to a set $B^1_{p,r}$ for some point $p \in N_{\varepsilon r}$, we need to update the value of $w_{\varepsilon r}(B^1_{p,r})$ as well as the value of $H_r[p]$. Each update in the max-heap $H_r$ can be done in $O(\log n)$ time using the standard max-heap operations. In the full version, we show that $2^{O(\mathrm{dim})\varepsilon^{-\mathrm{dim}}} \log \Delta$ updates happen to set $B^1$. Therefore, we need $2^{O(\mathrm{dim})\varepsilon^{-\mathrm{dim}}} \log \Delta \log n$ time to update the heaps.

**A few lemmas about our data structure.**    Now we mention a few useful lemmas about our data structure. The formal proofs can be found in the full version. The first lemma is about the distance of each point to its representative, which is bounded because of a geometric series argument.

**Lemma 2.1.** *Let $p \in P_t$ and $r \in R$. Then $d(p, rep_r(p)) \le 2r$. In particular, $d(p, rep_{\varepsilon r}(p)) \le 2\varepsilon r$, where $\varepsilon = 2^{\lfloor \log \varepsilon' \rfloor}$ for the error parameter $\varepsilon'$.*

We next discuss the size of close neighborhoods and extended neighborhoods.

**Lemma 2.2.** *Let $r \in R$ and $p \in N_{\varepsilon r}$. The sets $B^1_{p,r}$ and $B^3_{p,r}$ defined above are of size at most $2^{O(\mathrm{dim})}\varepsilon^{-\mathrm{dim}}$. In particular, the size is $O(\varepsilon^{-\mathrm{dim}})$ for a constant doubling dimension $\mathrm{dim}$.*

In the next lemma, we show that the update time is logarithmic in the spread ratio and the size of $P_t$.

**Lemma 2.3.** *Let $n$ be the size of point set $P_t$. Then after insertion/deletion of any point to $P_t$, we can update our data structure in time $2^{O(\mathrm{dim})}\varepsilon^{-2\,\mathrm{dim}} \log \Delta \log n$.*

Besides insertion and deletion updates, our data structure supports solution queries: in every time step, it can produce a solution in the form of a center set and a radius. We will discuss this in more detail and present an algorithm that outputs a $(3 + O(\varepsilon))$-approximation in time $\varepsilon^{-O(\mathrm{dim})}k \log n \log \log \Delta$ in the following section.

## 3   Our algorithm

**Overview**    For a given level $r$ in $R$ and a number of centers $k$, the main procedure MAXCOVERAGE produces a set of $k$ centers and a value *outliersWeight*. One run of this procedure is deemed successful if *outliersWeight* $\le z$ at the end. Indeed, we will later show that *outliersWeight* is an upper bound on the actual number of outliers in the clustering with centers $C$ and radius $(3+10\varepsilon)r$, which implies that this is a feasible solution. The algorithm FINDCENTERS finds one such level for which MAXCOVERAGE is successful. We also ensure that this level is chosen such that MAXCOVERAGE is not successful on the next lower level $r/2$, which guarantees that the radius of our solution does not deviate from the optimal solution by too much.

In our algorithm, we repeatedly find areas that subsume the most points. In order to do this efficiently, we maintain the collections of close neighborhoods and their weights in max-heaps. Fix a level $r \in R$. For $k$ iterations, the root node of the current heap corresponds to the next center of our solution. After picking it, we mark points in the extended neighborhood because our final approximate solution will cover these points and they should not contribute to the weight of other neighborhoods anymore.

As these points are now covered, we decrease the weights of heap elements representing points therein accordingly. These steps are performed by MAXCOVERAGE. Note that for maintenance reasons, one run of FINDCENTERS should leave the original heaps unchanged. We could make a copy of the heap before processing it in MAXCOVERAGE, but it is faster to work on the original heaps and just roll back all the updates done during MAXCOVERAGE at the end of the algorithm, as described in Lemma 3.8.

It remains to find a good value for $r$. By applying a binary search, we find a radius $r$ such that the value of *outliersWeight* outputted by MAXCOVERAGE($k$, $r$) is at most $z$, while the respective output on the lower level $r/2$ is more than $z$. Note that it is possible to find such a pair of subsequent radii via binary search although the search space might

---

**Procedure** MAXCOVERAGE($k$,$r$)

1   $C \leftarrow \emptyset$, Marked $\leftarrow \emptyset$, *outliersWeight* $\leftarrow n$
2   Let $H$ be a copy of $H_r$
3   **for** $i = 1$ *to* $k$ **do**
4      Let $c_i$ be the key with maximum value in heap $H$
5      $C \leftarrow C \cup \{c_i\}$
6      **for** *each* $y \in B^3_{c_i,r}$ *such that* $y \notin$ *Marked* **do**
7         Marked $\leftarrow$ Marked $\cup \{y\}$
8         **for** *each* $x \in B^1_{y,r}$ **do**
9            Decrease $H[x]$ by $w_{\varepsilon r}(y)$
10         *outliersWeight* $\leftarrow$ *outliersWeight* $-w_{\varepsilon r}(y)$

11   **return** $C$, *outliersWeight*

---

**Algorithm 1:** FINDCENTERS($k, z$)

**Output :** Centers $C$ of a $(3 + 10\varepsilon)\rho$-approximation for $(k, z)$-center
1   **if** MAXCOVERAGE($k, 0$). *outliersWeight* $\leq z$ **then**
2      **return** MAXCOVERAGE($k, 0$)
3   By performing a binary-search on $R$, find $\hat{r}$ such that MAXCOVERAGE($k, \hat{r}$). *outliersWeight* $\leq z$, and MAXCOVERAGE($k, \hat{r}/2$). *outliersWeight* $> z$ if $\hat{r} > 2$
4   **return** MAXCOVERAGE($k, \hat{r}$)

---

not be monotone. In the special case that there are at most $z$ points in total, we can just set the radius to 0. This is done in Algorithm 1 (FINDCENTERS($k, z$)).

**Analysis**    Next, we prove the approximation guarantee and time complexity of our algorithm.

Let $r$ and $k$ be fixed, and let $i \in [1, k]$ be an integer. For any point $y \in N_{\varepsilon r}$, we say $y$ is *marked* after the $i$-th iteration of the loop in Line 3 of MAXCOVERAGE, if $y \in \bigcup_{j=1}^{i} B^3_{c_j,r}$, and we say it is *unmarked* if it is not the case. For simplicity, we may refer to a point as marked or unmarked without specifying the iteration if it is clear from the context. We define set $U_i :=$ $\{y \in N_{\varepsilon r} \mid y \notin \bigcup_{j=1}^{i} B^3_{c_j,r}\}$ as the *set of unmarked points* after the $i$-th iteration. We next define $O_i := \{p \in P_t \mid \text{rep}_{\varepsilon r}(p) \notin \bigcup_{j=1}^{i} B^3_{c_j,r}\}$ as the set of points that are *considered outliers* after the $i$-th iteration of the loop. Equivalently, $O_i := \{p \in P_t \mid \text{rep}_{\varepsilon r}(p) \in U_i\}$. After the $i$-th iteration of the loop in Line 3, for each $y \in N_{\varepsilon r}$, $H_r[y]$ is the total weight of unmarked points in $B^1_{y,r}$.

**Lemma 3.1** (Invariant for heaps). *Let $k$ be an integer, $0 \leq i \leq k$, and $r \in R$. Then for any $y \in N_{\varepsilon r}$ it holds $H_r[y] = w_{\varepsilon r}(B^1_{y,r} \cap U_i)$ after the $i$-th iteration.*

FINDCENTERS($k, z$) returns a solution when *outliersWeight* $\leq z$. To show that this solution is feasible, it remains to prove that *outliersWeight* is an upper bound on the actual number of outliers of this solution.

**Lemma 3.2** (Feasible solution). *Consider a radius $r \in R$, and an integer $k$. Then MAXCOVERAGE($k, r$) returns $k$ centers and a value outliersWeight, such that the total number of points in $P_t$ that are not within distance $(3 + 10\varepsilon)r$ of these $k$ centers is at most outliersWeight.*

*Proof.* Let *outliersWeight* denote MAXCOVERAGE($k, r$). *outliersWeight*. We want to prove that MAXCOVERAGE($k, r$) returns $\{c_1, \ldots, c_k\}$ such that $|P_t \setminus \bigcup_{j \leq k} \text{Ball}(c_j, (3 + 10\varepsilon)r)| \leq$ *outliersWeight*. In every iteration $i$, *outliersWeight* is decreased by $w_{\varepsilon r}(y)$ for every $y \in B^3_{c_i,r} \setminus$ Marked. After processing $y$, we add it to Marked. This way, we ensure that no $y$ is charged twice. Therefore, by the end of the algorithm, *outliersWeight* is decreased by $w_{\varepsilon r}(\bigcup_{j \leq k} B^3_{c_j,r})$.

Hence, in the end, the value of *outliers Weight* is

$$outliersWeight = n - w_{\varepsilon r}\left(\bigcup_{j \le k} B^3_{c_j,r}\right)$$

$$= |P_t| - |\{p \in P_t \mid \operatorname{rep}_{\varepsilon r}(p) \in \bigcup_{j \le k} B^3_{c_j,r}\}| .$$

For all $p \in P_t$ , if $\operatorname{rep}_{\varepsilon r}(p) \in \bigcup_{j \le k} B^3_{c_j,r}$, then $d(c_j, \operatorname{rep}_{\varepsilon r}(p)) \le (3 + 8\varepsilon)r$ holds for at least one $j \le k$ by definition of $B^3_{c_j,r}$. Using Lemma 2.1, this implies $d(c_j, p) \le d(c_j, \operatorname{rep}_{\varepsilon r}(p)) + d(\operatorname{rep}_{\varepsilon r}(p), p) \le (3 + 8\varepsilon)r + 2\varepsilon r = (3 + 10\varepsilon)r$. Hence, $\{p \in P_t \mid \operatorname{rep}_{\varepsilon r}(p) \in \bigcup_{j \le k} B^3_{c_j,r}\} \subseteq \{p \in P_t \mid \exists j \le k \colon d(c_j, p) \le (3 + 10\varepsilon)r\}$ and therefore

$$outliersWeight \ge |P_t| - |\{p \in P_t \mid \exists j \le k \colon d(c_j, p) \le (3 + 10\varepsilon)r\}|$$

$$= |P_t| - |\bigcup_{j \le k} \operatorname{Ball}(c_j, (3 + 10\varepsilon)r) \cap P_t| . \qquad \square$$

Consider a fixed optimal solution. The following insight is crucial for our proof: In every iteration of $\text{MAXCOVERAGE}(k, r)$ for $r \ge \text{OPT}$, we select a ball that covers a sufficient number of unmarked points. To be precise, it exceeds the number of currently considered outliers in any optimal cluster.

**Lemma 3.3** (Invariant for picking $c_i$)**.** *Let $r \ge \text{OPT}$, where $r \in R$, and let $c^* \in P_t$ be a center of an optimal solution for $k$-center with $z$ outliers. Assume that we execute $\text{MAXCOVERAGE}(k, r)$. Then for any $i \in [1, k]$, the $c_i$ picked in Line 4 is such that $|\operatorname{Ball}(c^*, OPT) \cap O_{i-1}| \le w_{\varepsilon r}(B^1_{c_i,r} \cap U_{i-1})$.*

*Proof.* We define $\hat{c} := \operatorname{rep}_{\varepsilon r}(c^*)$, the representative of $c^*$ in $N_{\varepsilon r}$. Recall that $B^1_{\hat{c},r} = \{y \in N_{\varepsilon r} \mid d(\hat{c}, y) \le (1 + 4\varepsilon)r\}$. Lemma 3.1 states that $H[y] = w_{\varepsilon r}(B^1_{y,r} \cap U_{i-1})$ holds for all $y \in N_{\varepsilon r}$ after the $(i-1)$-th iteration. Since $c_i$ is on top of the max-heap $H$, we have $w_{\varepsilon r}(B^1_{\hat{c},r} \cap U_{i-1}) \le w_{\varepsilon r}(B^1_{c_i,r} \cap U_{i-1})$. Therefore, it remains to show that $|\operatorname{Ball}(c^*, \text{OPT}) \cap O_{i-1}| \le w_{\varepsilon r}(B^1_{\hat{c},r} \cap U_{i-1})$.

According to the definition of representative and weight function, we first have

$$|\operatorname{Ball}(c^*, \text{OPT}) \cap O_{i-1}| \le \sum_{y \in \operatorname{rep}_{\varepsilon r}(\operatorname{Ball}(c^*, \text{OPT}) \cap O_{i-1})} w_{\varepsilon r}(y) .$$

We next show that for any point $p \in \operatorname{Ball}(c^*, \text{OPT}) \cap P_t$, its representative on level $\varepsilon r$ lies in $B^1_{\hat{c},r}$, i.e. $\operatorname{rep}_{\varepsilon r}(p) \in B^1_{\hat{c},r}$. Let $p \in \operatorname{Ball}(c^*, \text{OPT}) \cap P_t$ be an arbitrary point, then $d(c^*, p) \le \text{OPT} \le r$. By the triangle inequality, $d(\hat{c}, \operatorname{rep}_{\varepsilon r}(p)) \le d(\hat{c}, c^*) + d(c^*, p) + d(p, \operatorname{rep}_{\varepsilon r}(p))$. Besides, Lemma 2.1 implies that $d(\hat{c}, c^*) \le 2\varepsilon r$ and $d(p, \operatorname{rep}_{\varepsilon r}(p)) \le 2\varepsilon r$. Putting everything together we have

$$d(\hat{c}, \operatorname{rep}_{\varepsilon r}(p)) \le d(\hat{c}, c^*) + d(c^*, p) + d(p, \operatorname{rep}_{\varepsilon r}(p)) \le 2\varepsilon r + r + 2\varepsilon r = (1 + 4\varepsilon)r .$$

Besides, $p \in O_{i-1}$ is equivalent to $\operatorname{rep}_{\varepsilon r}(p) \in U_{i-1}$. Therefore, $p \in \operatorname{Ball}(c^*, \text{OPT}) \cap O_{i-1}$ implies $\operatorname{rep}_{\varepsilon r}(p) \in B^1_{\hat{c},r} \cap U_{i-1}$, and we have $\operatorname{rep}_{\varepsilon r}(\operatorname{Ball}(c^*, \text{OPT}) \cap O_{i-1}) \subseteq B^1_{\hat{c},r} \cap U_{i-1}$. Thus,

$$\sum_{y \in \operatorname{rep}_{\varepsilon r}(\operatorname{Ball}(c^*, \text{OPT}) \cap O_{i-1})} w_{\varepsilon r}(y) \le \sum_{y \in B^1_{\hat{c},r} \cap U_{i-1}} w_{\varepsilon r}(y) = w_{\varepsilon r}(B^1_{\hat{c},r} \cap U_{i-1}) ,$$

which finishes the proof. $\qquad \square$

Utilizing Lemma 3.3, we can now show that the weight of the final output of $\text{MAXCOVERAGE}(k, r)$ with $r \ge \text{OPT}$ is at least the number of points covered by any optimal solution.

**Lemma 3.4** (Invariant for coverage weight)**.** *Let $C^*$ be the set of $k$ centers of an optimal solution for $k$-center with $z$ outliers on $P_t$, and let $\text{OPT}$ be the radius of this solution. Let $r \ge \text{OPT}$, where $r \in R$. Then,*

$$\left| \bigcup_{c^* \in C^*} \operatorname{Ball}(c^*, OPT) \cap P_t \right| \le w_{\varepsilon r}\left(\bigcup_{j=1}^{k} B^3_{c_j,r}\right)$$

*(recall that $B^3_{c_j,r} = \operatorname{Ball}(c_j, (3 + 8\varepsilon)r) \cap N_{\varepsilon r}$ and $w_{\varepsilon r}(B^3_{c_j,r}) = |\{y \in P_t \mid rep_{\varepsilon r}(y) \in B^3_{c_j,r}\}|$).*

*Proof.* To prove the lemma, we show that for any $i \in [0, k]$, there is a charging that satisfies the following conditions: 1) There is a set $\{c_1^*, \dots, c_i^*\} \subseteq C^*$ and a function that charges each point in $\bigcup_{j=1}^{i} \text{Ball}(c_j^*, OPT) \cap P_t$ to a point $y \in \bigcup_{j=1}^{i} B_{c_j, r}^3$, and 2) for any point $y \in \bigcup_{j=1}^{i} B_{c_j, r}^3$, at most $w_{\varepsilon r}(y)$ points from $\bigcup_{j=1}^{i} \text{Ball}(c_j^*, OPT) \cap P_t$ are mapped to $y$.

We prove this claim by induction on $i$. The base case $i = 0$ trivially holds. Assume that $i \geq 1$ and the induction hypothesis holds for $i - 1$. That is, there is a charging satisfying conditions 1 and 2 for $i - 1$. Then, we prove that these conditions hold for $i$.

To define center $c_i^*$, we consider two cases:

**Case 1:** there exists a center $c^* \in C^* \setminus \{c_1^*, \dots, c_{i-1}^*\}$ for which $B_{c_i, r}^1$ hits the set of representatives of $\text{Ball}(c^*, OPT)$. That is, we have $\text{rep}_{\varepsilon r}(\text{Ball}(c^*, OPT)) \cap B_{c_i, r}^1 \neq \emptyset$. We then let $c_i^* = c^*$. We map every point of $\text{Ball}(c^*, OPT)$ to its unique representative.

**Case 2:** the first case is not correct. Then, we let $c_i^*$ be any arbitrary point in $C^* \setminus \{c_1^*, \dots, c_{i-1}^*\}$. In this case, we charge the uncharged points of $\text{Ball}(c_i^*, OPT)$ to $B_{c_i, r}^1$.

We first consider Case 1. We charge each point $p \in \text{Ball}(c_i^*, OPT)$ to its representative $\text{rep}_{\varepsilon r}(p)$. We next prove the following claim:

*Claim.* Let $p \in \text{Ball}(c_i^*, OPT)$. If Case 1 happens, then for every point $q \in P_t$ that is charged to $\text{rep}_{\varepsilon r}(p)$, it holds that $\text{rep}_{\varepsilon r}(q) = \text{rep}_{\varepsilon r}(p)$.

*Proof.* For the sake of contradiction, assume that there is a point $q \in P_t$ such that $q$ is charged to $\text{rep}_{\varepsilon r}(p)$ and $\text{rep}_{\varepsilon r}(q) \neq \text{rep}_{\varepsilon r}(p)$. It means that at an iteration $j < i$, Case 2 happened and $q$ is charged to $\text{rep}_{\varepsilon r}(p)$. Since Case 2 happened at iteration $j$, we have $\text{rep}_{\varepsilon r}(\text{Ball}(c_i^*, OPT)) \cap B_{c_j, r}^1 = \emptyset$ and also $q$ is charged to a point in $B_{c_j, r}^1$. Adding it to the assumption that $q$ is charged to $\text{rep}_{\varepsilon r}(p)$ we have $\text{rep}_{\varepsilon r}(p) \in B_{c_j, r}^1$. Besides, $p \in \text{Ball}(c_i^*, OPT)$ and therefore we have $\text{rep}_{\varepsilon r}(p) \in \text{rep}_{\varepsilon r}(\text{Ball}(c_i^*, OPT))$. This implies that $\text{rep}_{\varepsilon r}(p) \in \text{rep}_{\varepsilon r}(\text{Ball}(c_i^*, OPT)) \cap B_{c_j, r}^1$, which is a contradiction to $\text{rep}_{\varepsilon r}(\text{Ball}(c_i^*, OPT)) \cap B_{c_j, r}^1 = \emptyset$. ◁

The claim implies that for each point $p \in \text{Ball}(c^*, OPT)$, at most $w_{\varepsilon r}(\text{rep}_{\varepsilon r}(p))$ points are charged to $\text{rep}_{\varepsilon r}(p)$. See Figure 2, where the left blue ball represents $B_{c_i, r}^1$, the right blue one indicates the area that points from $\text{rep}_{\varepsilon r}(\text{Ball}(c^*, OPT))$ can lie in, and the orange circle represents $\text{Ball}(c^*, OPT)$. For each point $p \in \text{Ball}(c^*, OPT)$, we have $\text{rep}_{\varepsilon r}(p) \in B_{c_i, r}^3$: Let $q \in \text{Ball}(c^*, OPT)$ be such that $\text{rep}_{\varepsilon r}(q) \in \text{rep}_{\varepsilon r}(\text{Ball}(c^*, OPT)) \cap B_{c_i, r}^1$. The distance of $c_i$ to any representative of a point $p \in \text{Ball}(c^*, OPT)$ is

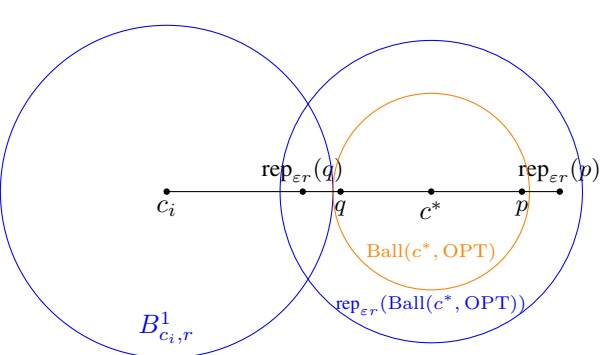

Figure 2: Illustration of the claim's consequences.

$$d(c_i, \text{rep}_{\varepsilon r}(p)) \leq d(c_i, \text{rep}_{\varepsilon r}(q)) + d(\text{rep}_{\varepsilon r}(q), q) + d(q, c^*) + d(c^*, p) + d(p, \text{rep}_{\varepsilon r}(p))$$
$$\leq (1 + 4\varepsilon)r + 2\varepsilon r + 2\,\text{OPT} + 2\varepsilon r \leq (3 + 8\varepsilon)r,$$

implying that $\text{rep}_{\varepsilon r}(p) \in B_{c_i, r}^3$.

For Case 2: note that if $p \in P_t \setminus O_{i-1}$, then by definition of $O_{i-1}$, we have $\text{rep}_{\varepsilon r}(p) \notin U_{i-1}$, and hence $\text{rep}_{\varepsilon r}(p) \in \cup_{j=1}^{i-1} B_{c_j, r}^1$ by definition of $U_{i-1}$. So, by induction hypothesis, $p$ is already charged to $\text{rep}_{\varepsilon r}(p)$. Therefore, we only need to charge each point $p \in \text{Ball}(c^*, OPT) \cap O_{i-1}$. By Lemma 3.3 we have $|\text{Ball}(c^*, OPT) \cap O_{i-1}| \leq w_{\varepsilon r}(B_{c_i, r}^1 \cap U_{i-1})$. Also, we only charged to

the points in $\cup_{j=1}^{i-1} B^3$, which means that no point is charged a point in $U_{i-1}$. Therefore, points in $\mathrm{Ball}(c^*, OPT) \cap O_{i-1}$ can be charged to points in $B^1_{c_i,r} \cap U_{i-1}$. $\square$

**Lemma 3.5.** *Let $r \in R$ and $OPT$ be the optimal radius for $k$-center clustering of $P_t$ with $z$ outliers. Then if $r \geq OPT$, the value outliersWeight returned by MAXCOVERAGE$(k, r)$ is at most $z$.*

*Proof.* Let $C = \{c_1, \ldots, c_k\}$ be the set of centers returned by MAXCOVERAGE$(k, r)$ and $C^*$ be the set of centers of an optimal $k$-center clustering of $P_t$ with $z$ outliers. By the same reasoning as in the proof of Lemma 3.2, it is

$$outliersWeight = n - w_{\varepsilon r}(\cup_{i=1}^{k} B^3_{c_i,r}) \ .$$

Besides, Lemma 3.4 states that $w_{\varepsilon r}(\cup_{i=1}^{k} B^3_{c_i,r}) \geq |\cup_{c^* \in C^*} \mathrm{Ball}(c^*, OPT) \cap P_t|$. Moreover, as $C^*$ are the centers of an optimal solution, $n - |\cup_{c^* \in C^*} \mathrm{Ball}(c^*, OPT) \cap P_t| \leq z$. Putting everything together, we have

$$outliersWeight \leq n - w_{\varepsilon r}(\cup_{i=1}^{k} B^3_{c_i,r}) \leq n - |\cup_{c^* \in C^*} \mathrm{Ball}(c^*, OPT) \cap P_t| \leq z \ ,$$

which finishes the proof. $\square$

Lemma 3.2 and Lemma 3.5 together imply that the solution returned by Algorithm 1 is feasible. Further, it can be shown that the level $\hat{r}$ at which FINDCENTERS$(k, z)$ becomes successful, fulfills $\hat{r} \leq r^*$. Together, this implies the following statement.

**Lemma 3.6** (Approximation guarantee). *Let $OPT > 0$ be the optimal radius for the $k$-center clustering of $P_t$ with $z$ outliers, and let $r^*$ be the minimum number in $R$ such that $r^* \geq OPT$. Then FINDCENTERS$(k, z)$ returns a $(3 + 10\varepsilon)\rho$-approximate solution for $k$-center clustering problem with $z$ outliers, where $\rho = \frac{r^*}{OPT}$.*

For $\rho$ defined in Lemma 3.6, $1 \leq \rho < 2$ holds according to the definition of $R$. It immediately leads to a $(6 + O(\varepsilon))$-approximation algorithm. Next in Lemma 3.7, we show how to get $\rho \leq (1 + \varepsilon)$, which improves the approximation ratio of our algorithm to $3 + O(\varepsilon)$. The idea is to run parallel instances of our data structure with different values for parameter $\beta$, so that always $OPT \leq r < (1+\varepsilon)OPT$ holds for a $r \in R$ in one of the instances.

**Lemma 3.7** (Optimizing the approximation ratio). *Let $OPT > 0$, and let $\varepsilon > 0$ be fixed. We define $m := \lceil 1/\log_2 (1 + \varepsilon) \rceil$. Suppose we have $m$ parallel instances of our data structure with parameter $\beta = 2^{i/m-1}$ for the $i$-th instance. Then in at least one of the instances, we find a $(3 + O(\varepsilon))$-approximation solution for $k$-center clustering with $z$ outliers by calling FINDCENTERS$(k, z)$.*

The procedure MAXCOVERAGE$(k, r)$ needs to temporarily edit the heap $H_r$. This can be done by working on a copy of the heap, as indicated in line 2. The following result describes how this can be done faster, without actually copying the heap. The idea is to work in place and roll back all changes in the end. The formal proof is given in the full version.

**Lemma 3.8** (Imitating of copying $H_r$). *Let $u$ be the number of times that MAXCOVERAGE$(k, r)$ updates heap $H$. Then copying heap $H_r$ in Line 2, as well as all these $u$ updates, can be handled in total time $O(u \log n)$.*

By using Lemma 3.8 and Lemma 2.2, we get the query time in Lemma 3.9.

**Lemma 3.9** (Query time). *Let $k$ and $z$ be two given parameters. Then FINDCENTERS$(k, z)$ described in Algorithm 1 has a time complexity of $2^{O(\dim)} \varepsilon^{-2\dim} k \log n \log \log \Delta$.*

# 4 Conlusion

We developed a data structure for the fully dynamic $k$-center with $z$ outliers problem in metrics of bounded doubling dimension. As compared to other works, the algorithm exhibits an improved query time while achieving the currently best-known approximation ratio of $3 + \varepsilon$. The query time is optimal with respect to the dependency on $k$. Although the exponential dependency on the doubling dimension in the running times seems necessary, it is actually not clear if this could be improved to tackle even wider classes of metric spaces. This would be an interesting aspect for future work. However, Chan et al. [4] show that an amortized running time of $\Omega(z)$ would be needed in general metric spaces.

# 5    Acknowledgements

The authors would like to thank the anonymous reviewers for their insightful comments. Annika Hennes' and Melanie Schmidt's research was funded by the Deutsche Forschungsgemeinschaft (DFG, German Research Foundation) – project number 456558332.

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

# A    Maintaining the data structure

As we explained in Section 2, our data structure consists of $O(\log \Delta)$ levels $R := \{0\} \cup \{\beta \cdot 2^\ell \mid \ell \in \mathbb{Z}, \ 0 \le \ell \le \lceil \log_2 \Delta \rceil + 1\}$, where $\frac{1}{2} < \beta \le 1$ is a given parameter. Similar to Krauthgamer and Lee [15], on each level $r$, we maintain net $N_r$, which is an $r$-net of $N_{r/2}$. In this section, we explain how to maintain our data structure after each insert/delete update, and analyze the update time.

For every point $p \in N_r$, we maintain the sets of all its parents and all its children. We start by bounding the number of children and parents of each point. Recall that $p \in N_r$ is called a parent of $q \in N_{r/2}$ if $d(p,q) \le r$. We first show that the number of children for each point is of size $2^{O(\dim)}$.

**Lemma A.1** (Number of children). *Let $p \in N_r$. The number of children of $p$ is at most $2^{2\dim}$.*

*Proof.* For every child $q \in N_{r/2}$ of $p$ it is $d(p,q) \le r$. Therefore, the set of children is covered by $\mathrm{Ball}(p, r)$. Using the definition of doubling dimension twice, it follows that the set of children can be covered by $2^{2\dim}$ balls of radius $r/4$. As any two points in $N_{r/2}$ have distance at least $r/2$, the claim follows. $\qquad\square$

Then, in a similar fashion, we can deduce the number of parents.

**Lemma A.2** (Number of parents). *Let $p \in N_r$. The number of parents of $p$ is bounded by $2^{O(\dim)}$.*

## A.1    Finding the neighbours

In this section, we explain FindNeighbours, which takes as input a value $\gamma \ge 2$ and a point $p \in P_t$ and returns for every $r \in R \setminus \{0\}$ the set of neighbors of $p$ within distance $\gamma r$. Later, we use this procedure to maintain our data structure after insertions and deletions.

---

**Procedure** FINDNEIGHBOURS$(p, \gamma)$

---
1  $r_{\max} := \beta \cdot 2^{\lceil \log_2 \Delta \rceil}$
2  $r \leftarrow r_{\max}$
3  $S_r \leftarrow N_r$
4  **while** $r > \beta$ **do**
5  $\quad$ $r \leftarrow r/2$
6  $\quad$ $S_r \leftarrow \emptyset$
7  $\quad$ **for** *all* $s \in S_{2r}$ **do**
8  $\quad\quad$ **for** *all* children $q \in N_r$ of $s$ **do**
9  $\quad\quad\quad$ **if** $d(p,q) \le \gamma \cdot r$ **then**
10 $\quad\quad\quad\quad$ $S_r \leftarrow S_r \cup \{q\}$

11 **return** $S_\beta, S_{2\beta}, S_{4\beta}, S_{8\beta}, \ldots, S_{r_{max}}$

---

**Lemma A.3.** *Let $p \in P$, and $\gamma \ge 2$. Then, for all $r \in R \setminus \{0\}$, set $S_r$ that we compute in FindNeighbours$(p, \gamma)$ consists of all elements in $N_r$ within distance $\gamma \cdot r$ of $p$.*

*Proof.* We show the statement via induction on $r$. For $r = r_{\max} = \beta \cdot 2^{\lceil \log \Delta \rceil}$, line 3 implies $S_r = N_r$ and therefore the statement is trivially fulfilled. Now assume the statement holds for $2r$, where $r < r_{\max}$. Consider $x \in N_r$ such that $d(p,x) \le \gamma r$. Then, $d(p, \mathrm{rep}_{2r}(x)) \le d(p,x) + d(x, \mathrm{rep}_{2r}(x)) \le \gamma r + 2r \le 2\gamma r$, hence, by induction hypothesis, $\mathrm{rep}_{2r}(x) \in S_{2r}$. This means, $x$ is a child of an element in $S_{2r}$, and the condition of the for loop in line 8 holds. As further $d(p,x) \le \gamma r$ by assumption, $x$ is added to $S_r$ in line 10. $\qquad\square$

Next, we bound the size of set $S_r$ in Lemma A.4.

**Lemma A.4.** *Let $r \in R$. Then, set $S_r$ computed in FindNeighbours$(p, \gamma)$ is of size $\gamma^{\dim} 2^{O(\dim)}$.*

*Proof.* The condition in Line 9 of FindNeighbours implies that any point $x \in S_r$ is within distance $\gamma \cdot r$ of $p$. Hence, $S_r$ is covered by $\mathrm{Ball}(p, \gamma \cdot r)$. Besides, $S_r \subseteq N_r$, which means that the distance between any two points in $S_r$ is more than $r$. By applying the definition of doubling dimension, it

follows that $\text{Ball}(p, \gamma \cdot r)$ can be covered by $\gamma^{\dim} \cdot 2^{O(\dim)}$ balls of radius $r/2$. Adding it to $S_r \subseteq N_r$ implies that each of these balls of radius $r/2$ contains at most one point from $S_r$. Therefore, $S_r$ is of size $\gamma^{\dim} \cdot 2^{O(\dim)}$. $\qquad\square$

**Lemma A.5.** *Let $p \in P$ and $\gamma \geq 2$ be given. Then, the procedure FindNeighbours$(p, \gamma)$ runs in time $2^{O(\dim)}\gamma^{\dim} \log \Delta$.*

*Proof.* The while loop in Line 4 iterates backward through all the $O(\log \Delta)$ levels. On every level, all children of all $s \in S_{2r}$ are examined. By Lemma A.4, $|S_{2r}| \in \gamma^{\dim}2^{O(\dim)}$ and by Lemma A.1, every $s \in |S_{2r}|$ has $2^{2\dim}$ children. Multiplying these terms yields the claim. $\qquad\square$

### A.2  Insertion

In this section, we show how to maintain our data structure after the insertion of point $p$. As described in Algorithm 2, we start by calling FindNeighbours, which goes through the levels from top to bottom (high to low) and recursively computes set $S_r$ for each level $r$. Lemma A.3 implies that for any point $q \in N_r$, if $d(p, q) \leq r$, then $S_r$ contains $q$. This guarantees that later we can find all parents and children of $p$ using these $S_r$ sets. Next, we go through the levels from bottom to top to find the lowest level $\hat{r}$, in which we can find a representative for $p$, or in other words, there is a $q \in N_{\hat{r}}$ within distance $\hat{r}$ of $p$. Then, we add $p$ to all levels lower than $\hat{r}$. For levels $r > \hat{r}$, we do not need to add $p$ to $N_r$ because there is already a point in level $r$ that can represent $p$ by the definition of $\hat{r}$. We determine this representative recursively according to $\text{rep}_{\hat{r}}(p)$. Finally, we increase the weight for the respective representative of $p$ in each level by one.

---

**Algorithm 2:** INSERT$(p)$

---

1 $r_{\max} := \beta \cdot 2^{\lceil \log_2 \Delta \rceil}$
2 $S_\beta, S_{2\beta}, S_{4\beta}, S_{8\beta}, \ldots, S_{r_{max}} \leftarrow$ FindNeighbours$(p, 2)$
3 // finding the lowest level $\hat{r}$ that can have a representative for $p$
4 $r \leftarrow \beta$, $\hat{r} \leftarrow 2r_{\max}$
5 **while** $r \leq r_{\max}$ **do**
6     **if** *there is a point $s \in S_r$ such that $d(p, s) \leq r$* **then**
7         $\hat{r} \leftarrow r$
8         $\text{rep}_{\hat{r}}(p) \leftarrow s$
9         **for all** $s \in S_r$ such that $d(p, s) \leq r$ **do**
10             Save $s$ as a parent of $p$ and save $p$ as a child of $s$
11         break
12     $r \leftarrow 2r$
13 // adding $p$ to all levels lower than $\hat{r}$
14 **for all** $r \in R$ such that $r < \hat{r}$ **do**
15     $N_r \leftarrow N_r \cup \{p\}$
16     Save $p \in N_r$ as a parent of $p \in N_{r/2}$ and save $p \in N_{r/2}$ as a child of $p \in N_r$
17     $\text{rep}_r(p) \leftarrow p$
18 UPDATENEIGHBOURHOODADD$(p, \hat{r})$
19 // finding the representative of $p$ in all levels upper than $\hat{r}$
20 $r \leftarrow 2\hat{r}$
21 **while** $r \leq r_{\max}$ **do**
22     $\text{rep}_r(p) \leftarrow \text{rep}_r(\text{rep}_{r/2}(p))$
23     $r \leftarrow 2r$
24 // updating the weight function
25 **for all** $r \in R$ **do**
26     $w_r(\text{rep}_r(p)) \leftarrow w_r(\text{rep}_r(p)) + 1$

---

After inserting the point $p$ to $N_{\varepsilon r}$, we need to compute $B^1_{p,r}$ and $B^3_{p,r}$. Besides, for any point $q \in B^1_{p,r}$ ($q \in B^3_{p,r}$ similarly), we need to add $p$ to $B^1_{q,r}$ ($B^3_{q,r}$ similarly). We perform this operation

---

**Procedure** UPDATENEIGHBOURHOODADD($p, \hat{r}$): a procedure to update neighbourhoods after adding $p$ to all levels below $N_{\hat{r}}$

---

1   $r_{\max} := \beta \cdot 2^{\lceil \log_2 \Delta \rceil}$
2   $T_\beta, T_{2\beta}, T_{4\beta}, T_{8\beta}, \ldots, T_{r_{max}} \leftarrow \text{FindNeighbours}(p, (3 + 8\varepsilon)/\varepsilon)$
3   // updating neighborhoods $B^1$ and $B^3$
4   **for all** $r' \in R$ such that $r' < \hat{r}$ and $r'/\varepsilon \le r_{\max}$ **do**
5      $r := r'/\varepsilon$                                          $\triangleright \ r' = \varepsilon r$
6      $B^3_{p,r} \leftarrow \emptyset, \ B^1_{p,r} \leftarrow \emptyset$
7      **for all** $q \in T_{r'}$ **do**
8          $B^3_{p,r} \leftarrow B^3_{p,r} \cup \{q\}, \ B^3_{q,r} \leftarrow B^3_{q,r} \cup \{p\}$         $\triangleright \ d(p,q) \le (3 + 8\varepsilon)r$
9          **if** $d(p,q) \le (1 + 4\varepsilon)r$ **then**
10              $B^1_{p,r} \leftarrow B^1_{p,r} \cup \{q\}, \ B^1_{q,r} \leftarrow B^1_{q,r} \cup \{p\}$

---

in Procedure UpdateNeighbourhoodAdd. In Lemma A.6, we analyze the correctness and update time of this procedure.

**Lemma A.6.** *Let $p$ be a point to be inserted and $\hat{r}$ as in the algorithm after execution of the while loop in Line 5. Then,* UPDATENEIGHBOURHOODADD$(p, \hat{r})$ *that we invoke in Line 18 of Algorithm 2, finds $B^1_{p,r}$ and $B^3_{p,r}$ for any $r \in R$ such that $\varepsilon r < \hat{r}$. Besides, the total update time of this procedure is $2^{O(\dim)} \varepsilon^{-\dim} \log \Delta$.*

*Proof.* By Lemma A.3, $T_{r'}$ contains all points of $N_{r'}$ within distance $(3 + 8\varepsilon)r'/\varepsilon$ of $p$. Then if we define $r := r'/\varepsilon$, it holds that $T_{r'}$ contains all points of $N_{\varepsilon r}$ within distance $(3 + 8\varepsilon)r'/\varepsilon = (3 + 8\varepsilon)r$ of $p$. This means that we can find $B^3_{p,r}$ by exploring $T_{r'}$, which we do in Line 8 of the procedure. By restricting to points $q$ in $T_{r'}$ such that $d(p,q) \le (1 + 4\varepsilon)r$, we find all the points that belong to $B^1_{p,r}$.

We next discuss the running time. Lemma A.5 states that the running time of FindNeighbours$(p, (3 + 8\varepsilon)/\varepsilon)$ is $2^{O(\dim)} \varepsilon^{-\dim} \log \Delta$. Then, we iterate over $T_{r'}$ for at most $O(\log \Delta)$ values of $r'$, which takes $O(|T_{r'}|)$ for each iteration. Lemma A.4 shows that $T_{r'}$ is of size $((3 + 8\varepsilon)/\varepsilon)^{\dim} 2^{O(\dim)} = 2^{O(\dim)} \varepsilon^{-\dim}$. Therefore, the total running time of these $O(\log \Delta)$ iterations is also $2^{O(\dim)} \varepsilon^{-\dim} \log \Delta$, which finishes the proof. $\square$

**Updating the heaps.** To avoid redundancy, we do not include updating Heaps $H_r$ in the pseudocodes. However, it can be easily handled. Note that neighborhoods $B^1_{x,r}$ and $B^1_{y,r}$ are balls of the same size. Therefore, $y \in B^1_{x,r}$ is equivalent to $x \in B^1_{y,r}$. Hence, whenever the weight of a point $x$ in $N_r$ is changed, it is enough to update the weight of all points in $B^1_{x,r}$ and their corresponding element in heap $H_r$. The size of $B^1_{x,r}$ is $2^{O(\dim)} \varepsilon^{-\dim}$ by Lemma 2.2, and we can do each heap update in $O(\log n)$ time. Therefore, an extra $2^{O(\dim)} \varepsilon^{-\dim} \log n$ factor will be multiplied by the running time of Algorithm 2 due to maintaining the heaps.

**Lemma A.7.** *Let $n$ be the size of point set $P_t$. Then after insertion of any point to $P_t$, we can update our data structure in time $2^{O(\dim)} \varepsilon^{-2\dim} \log \Delta \log n$.*

*Proof.* As we discussed above, a $2^{O(\dim)} \varepsilon^{-\dim} \log n$ factor will be multiplied by the running time of Algorithm 2 due to maintaining the heaps. Therefore, it is enough to show that the running time of Algorithm 2 is $2^{O(\dim)} \varepsilon^{-\dim} \log \Delta$.

In Algorithm 2, we first call FindNeighbours$(p, 2)$, which takes $2^{O(\dim)} \log \Delta$ according to Lemma A.5. To find $\hat{r}$, we next iterate over $S_r$ for at most $O(\log \Delta)$ values of $r$. Then, we add $p$ to all levels lower than $\hat{r}$, which takes time $\log \Delta$. We next invoke UpdateNeighbourhoodAdd$(p, \hat{r})$, which is done in time $2^{O(\dim)} \varepsilon^{-\dim} \log \Delta$ according to Lemma A.6. This takes $2^{O(\dim)} \log \Delta$, since $S_r$ is of size $2^{O}(\dim)$ by Lemma A.4. Finally, we update the representatives of $p$ and their weights, which is done in time $O(\log \Delta)$. Putting everything together proves that the running time of Algorithm 2 is $2^{O(\dim)} \varepsilon^{-\dim} \log \Delta$, which finishes the proof. $\square$

## A.3 Deletion

In this section, we explain how to maintain our data structure after the deletion of point $p$. As described in Algorithm 3, we first produce sets $S_r$ and $T_r$ for all $r \in R$ that are used for updating parents and children as well as neighborhoods. We then iterate over the levels $r \in R$ from bottom to top. During the algorithm, we may elevate some points from $N_{r/2} \setminus N_r$ to $N_r$ to maintain our data structure. We refer to the set of these points that are elevated to $N_r$ as $E_r$. After changing the representative of a point, we need to change the weights of its previous representative in the next level as well as the indirect previous representatives in the upper levels. To make this process of updating weights efficient, we define the set $D_r$ and a function $\delta_r : D_r \to \mathbb{Z}$ for each $r \in R$. The set $D_r$ refers to the set of points $q \in N_r$ whose weight should be updated by the value $\delta_r[q]$.

After deleting $p$, we need to find a new representative for any point $q \in N_{r/2}$ and also reduce $w_{r/2}(q)$ from its previous indirect representatives in the upper levels. To do this, we decrease $\delta_{2r}[\text{rep}_{2r}(p)]$ by $w_{r/2}(q)$. After deleting $p$ from $N_r$, any point $q \in N_{r/2}$ that was represented by $p$ in $N_r$ would not have any representative. Besides, we need to find a representative in $N_r$ for the points in $E_{r/2}$. Let $q \in N_{r/2}$ be a point such that $\text{rep}_r(q) = p$ or $q \in E_{r/2}$. To find a new representative for $q$ in $N_r$, we consider two cases: either $q$ has a parent in $N_r \setminus \{p\}$ or $p$ is the only parent of $q$ in $N_r$.

For the first case, we choose a parent $x \in N_r \setminus \{p\}$ of $q$ as its representative in $N_r$. Then, we increase $w_r(x)$ and $\delta_{2r}[\text{rep}_{2r}(x)]$ by $w_{r/2}(q)$ to update the weight of all points that represent $q$ in higher levels directly or indirectly. For the second case, we have to elevate $q$ to $N_r$. After elevating $q$ to $N_r$, we need to find its parents and children. To find the children, it is enough to explore $S_{r/2}$. It is because if $s \in N_{r/2}$ is a child of $q \in N_r$, then $d(q,s) \le r$. Adding it to $d(p,q) \le r$, we have $d(p,s) \le 2r = 4(r/2)$ by triangle inequality. Therefore, $s \in S_{r/2}$ and it is enough to explore $S_{r/2}$. With a similar argument, it is enough to explore $S_{2r}$ to find the parents of $q$ in $N_{2r}$.

We next call UpdateNeighbourhoodElevate$(q, r, T_r)$. In Lemma A.9, we show that this procedure can compute $B^1_{q,r/\varepsilon}$ and $B^3_{q,r/\varepsilon}$ by exploring $T_r$.

At the end of each iteration, we call UPDATENEIGHBOURHOODREMOVE$(p, r)$. This procedure deletes $p$ from all the neighborhoods of level $r$ that contain $p$. Note that for $i \in \{1, 3\}$, $p \in B^i_{x,r}$ and $x \in B^i_{p,r}$ are equivalent. Hence, it is enough to iterate over all $x \in B^i_{p,r}$ and delete $p$ from $B^i_{x,r}$ for $i = \{1, 3\}$.

To show that UpdateNeighbourhoodElevate computes the neighborhoods in Lemma A.9, we first prove Lemma A.8, which is about the distance of the points in $E_r$ to $p$.

**Lemma A.8.** *Let $r \in R$, and let $E_r$ be the set of elevated points in Algorithm 3. Then, for any point $q \in E_r$ we have $d(p,q) \le r$.*

*Proof.* Let $\hat{r}$ be the smallest level in $R$ such that $q$ is elevated to it. It means that $q$ is a child of $p \in N_{\hat{r}}$, and therefore, $d(p,q) \le \hat{r}$. Besides, we have $\hat{r} \le r$, which implies $d(p,q) \le r$. $\square$

We next call UPDATENEIGHBOURHOODELEVATE to create $B^1_{q,r}$ and $B^3_{q,r}$, as well as to add $q$ to the neighborhoods in level $r$ that must contain $q$.

**Lemma A.9.** *Let $r' \in R$, $q \in E_{r'}$ and $T_{r'}$ as constructed in Line 3 of Algorithm 3. We define $r := r'/\varepsilon$. The procedure UpdateNeighbourhoodElevate$(q, r', T_{r'})$ computes $B^1_{q,r}$ and $B^3_{q,r}$ and updates the neighborhoods $B^1_{q',r}$ (and $B^3_{q',r}$) for all $q' \in B^1_{q,r}$ ($q' \in B^3_{q,r}$) by adding $q$ to them. Besides, this procedure runs in time $O(\varepsilon^{-\dim} 2^{O(\dim)})$.*

*Proof.* Let $p$ be the point to be deleted when calling Algorithm 3. $q \in E_{r'}$ implies that either $p$ was the representative of $q$ in $N_{r'}$, or $q \in E_{r'/2}$. If $p$ was the representative of $q$ in $N_{r'}$, we have $d(p,q) \le r'$. Otherwise, if $q \in E_{r'/2}$, Lemma A.8 states that $d(p,q) \le r'/2 \le r'$. Therefore, $d(p,q) \le r'$ always holds. By Lemma A.3, $T_{r'}$ contains all points $x \in N_{r'}$ such that $d(p,x) \le (1 + (3 + 8\varepsilon)/\varepsilon)r' = r' + (3 + 8\varepsilon)r$, for $r = r'/\varepsilon$. Let $q' \in N_{r'}$ be a point such that $d(q,q') \le (3 + 8\varepsilon)r$. We claim that we add $q'$ to $B^3_{q,r}$. To do this, it is enough to show that $q' \in T_{r'}$. As we mentioned, $d(p,q) \le r'$ and $d(q,q') \le (3 + 8\varepsilon)r$. Then by the triangle inequality we have $d(p,q') \le d(p,q) + d(q,q') \le r' + (3 + 8\varepsilon)r$, which means that $q' \in T_{r'}$ and it is enough to explore $T_{r'}$ for computing $B^3_{q,r}$. Note that the same argument also works for $B^1_{q,r}$.

**Algorithm 3:** DELETE($p$)

---

**1** $r_{\max} := 2^{\lceil \log_2 \Delta \rceil}$

**2** $S_\beta, S_{2\beta}, S_{4\beta}, S_{8\beta}, \ldots, S_{r_{max}} \leftarrow$ FindNeighbours($p, 4$)

**3** $T_\beta, T_{2\beta}, T_{4\beta}, T_{8\beta}, \ldots, T_{r_{max}} \leftarrow$ FindNeighbours($p, 1 + (3 + 8\varepsilon)/\varepsilon$)

**4** $E_{\beta/2} \leftarrow \emptyset, \ D_\beta \leftarrow \emptyset, \ \delta_\beta \leftarrow \emptyset$

**5** $r \leftarrow \beta$

**6 while** $r \leq r_{\max}$ **do**

**7**     $E_r \leftarrow \emptyset$

**8**     $D_{2r} \leftarrow \emptyset \ \ \delta_{2r} \leftarrow \emptyset$

**9**     // updating the weights of the points in the level

**10**     $w_r(\text{rep}_r(p)) \leftarrow w_r(\text{rep}_r(p)) - 1$

**11**     **for all** $q \in D_r$ **do**

**12**        $w_r(q) \leftarrow w_r(q) + \delta_r[q]$

**13**        $D_{2r}, \delta_{2r} \leftarrow$ UPDATEWEIGHT($D_{2r}, \delta_{2r}, \text{rep}_{2r}(q), \delta_r[q]$)

**14**     // updating the level

**15**     **for all** $q \in N_{r/2} \setminus \{p\}$ *that* $rep_r(q) = p$ *and* **all** $q \in E_{r/2}$ **do**

**16**        **if** $rep_r(q) = p$ **then**

**17**           $D_{2r}, \delta_{2r} \leftarrow$ UPDATEWEIGHT($D_{2r}, \delta_{2r}, \text{rep}_{2r}(p), -w_{r/2}(q)$)

**18**        // finding a new representative in $N_r$ for $q$

**19**        **if** $q$ *has a parent* $x$ *in* $N_r \setminus \{p\}$ **then**

**20**           $\text{rep}_r(q) \leftarrow x$

**21**           $w_r(x) \leftarrow w_r(x) + w_{r/2}(q)$

**22**           $D_{2r}, \delta_{2r} \leftarrow$ UPDATEWEIGHT($D_{2r}, \delta_{2r}, \text{rep}_{2r}(x), w_{r/2}(q)$)

**23**        **else**

**24**           // adding $q$ to $N_r$

**25**           $N_r \leftarrow N_r \cup \{q\}, \ E_r \leftarrow E_r \cup \{q\}$

**26**           $\text{rep}_r(q) \leftarrow q \ , \ w_r(q) \leftarrow w_{r/2}(q)$

**27**           **if** $d(q, p) \leq 4r$ **then**

**28**              $S_r \leftarrow S_r \cup \{q\}$

**29**           **if** $d(q, p) \leq (1 + (3 + 8\varepsilon)/\varepsilon) \cdot r$ **then**

**30**              $T_r \leftarrow T_r \cup \{q\}$

**31**           // finding all children of $q$ by exploring $S_{r/2}$

**32**           **for all** $s \in S_{r/2}$ **do**

**33**              **if** $d(q, s) \leq r$ **then**

**34**                 Save $q$ as a parent of $s$, and $s$ as a child of $p$

**35**           // finding all parents of $q$ in $N_{2r}$ by exploring $S_{2r}$

**36**           **for all** $s \in S_{2r}$ **do**

**37**              **if** $d(q, s) \leq 2r$ **then**

**38**                 Save $s$ as a parent of $q$, and $q$ as a child of $s$

**39**           UpdateNeighbourhoodElevate($q, r, T_r$)

**40**     $N_r \leftarrow N_r \setminus \{p\}$

**41**     $r \leftarrow 2r$

**42**     UPDATENEIGHBOURHOODREMOVE($p, r$)

**43** Delete all the data (such as parents, children, weight, and representative) corresponding to $p$

---

Further, $B_{q,r}^3$ (and $B_{q,r}^1$) contains all points whose neighborhoods need to be updated to include $q$. The procedure iterates through all $q' \in T_{r'}$, so the running time is $O(|T_{r'}|) = O(\varepsilon^{-\dim} 2^{O(\dim)})$. $\square$

---

**Procedure** UPDATEWEIGHT($D, \delta, q, \omega$)

1  **if** $q \notin D$ **then**
2     $D \leftarrow D \cup \{q\}$
3     $\delta[q] \leftarrow 0$
4  $\delta[q] \leftarrow \delta[q] + \omega$
5  **return** $C, \delta$

---

**Procedure** UPDATENEIGHBOURHOODELEVATE($q, r', T_{r'}$): a procedure to update neighbourhoods after adding $q$ to $N_{r'}$

1  $r := r'/\varepsilon$                                             $\triangleright$ $r' = \varepsilon r$
2  **if** $r > \beta \cdot 2^{\lceil \log_2 \Delta \rceil}$ **then**
3     **return**
4  $B_{q,r}^3 \leftarrow \emptyset, \ B_{q,r}^1 \leftarrow \emptyset$
5  **for** *all* $q' \in T_{r'}$ **do**
6     **if** $d(q, q') \leq (3 + 8\varepsilon)r$ **then**
7         $B_{q,r}^3 \leftarrow B_{q,r}^3 \cup \{q\}, \ B_{q',r}^3 \leftarrow B_{q',r}^3 \cup \{p\}$
8     **if** $d(q, q') \leq (1 + 4\varepsilon)r$ **then**
9         $B_{q,r}^1 \leftarrow B_{q,r}^1 \cup \{q'\}, \ B_{q',r}^1 \leftarrow B_{q',r}^1 \cup \{q\}$

---

**Procedure** UPDATENEIGHBOURHOODREMOVE($p, r'$): a procedure to update neighbourhoods after removing $p$ from $N_{r'}$

1  $r := r'/\varepsilon$                                             $\triangleright$ $r' = \varepsilon r$
2  **if** $r > \beta \cdot 2^{\lceil \log_2 \Delta \rceil}$ *or* $p \notin N_{r'}$ **then**
3     **return**
4  **for** *all* $q \in B_{p,r}^3$ **do**
5     $B_{q,r}^3 \leftarrow B_{q,r}^3 \setminus \{p\}$
6  **for** *all* $q \in B_{p,r}^1$ **do**
7     $B_{q,r}^1 \leftarrow B_{q,r}^1 \setminus \{p\}$
8  Delete $B_{p,r}^3$ and $B_{p,r}^1$

---

To prove the update time of Algorithm 3, we first prove Lemma A.10 and Lemma A.11.

**Lemma A.10.** *Let $r \in R$. Then, the set $E_r$ in Algorithm 3 is of size $2^{O(\dim)}$.*

*Proof.* Lemma A.8 states that $d(p, q) \leq r$ holds for any $q \in E_r$. Moreover, $E_r \subseteq N_r$, which implies that the distance between any two points in $E_r$ is more than $r$ by the definition of $N_r$. Therefore, the definition of doubling dimension implies that $|E_r| \leq 2^{O(\dim)}$. $\square$

**Lemma A.11.** *Let $r \in R$. Then the set $D_r$ in Algorithm 3 is of size $2^{O(\dim)}$.*

*Proof.* To prove the lemma, we first claim that for any $r \in R$, if $y \in D_r$, then $d(p, y) \leq 2r$. We prove the claim by induction on $r$. For the base case, we consider $r = \beta$. Since $D_\beta = \emptyset$, the claim trivially holds for $r = \beta$. Now, we show that if the claim holds for $r \in R$, then it also holds for $2r$. Let $v$ be a point in $D_{2r}$. Then, $v$ is added to $D_{2r}$ in one of the Lines 13, 17, or 22 of Algorithm 3. If it is added in Line 13, it means that there is a point $q \in D_r$, such that $v = \text{rep}_{2r}(q)$. By the induction hypothesis, we have $d(p, q) \leq 2r$. Besides, we know that

$d(q, v) = d(q, \mathrm{rep}_{2r}(q)) \le 2r$. Hence, $d(p, v) \le 4r = 2(2r)$ holds by the triangle inequality. If $v$ is added to $D_{2r}$ in Line 17, we have $v = \mathrm{rep}_{2r}(p)$, and we know that $d(p, \mathrm{rep}_{2r}(p)) \le 2r$. Therefore, $d(p, v) = d(p, \mathrm{rep}_{2r}(p)) \le 2r \le 2(2r)$. Finally, we consider the case that $v$ is added to $D_{2r}$ in Line 22. In this case, $v = \mathrm{rep}_{2r}(x)$ for a point $x \in N_r$, and $x = \mathrm{rep}_r(q)$ for a point $q \in N_{r/2}$ such that either $q$ was a child of $p$ or $q \in E_{r/2}$. If $q$ was a child of $p$, then $d(p, q) \le r$, and if $q \in E_{r/2}$, then $d(p, q) \le r/2 \le r$ by Lemma A.8. Therefore, $d(p, q) \le r$ always holds. Besides, $d(q, x) \le r$, as $x$ is a parent of $q$. Moreover, we have $d(x, \mathrm{rep}_{2r}(x)) \le 2r$. Putting everything together we have

$$d(p, v) = d(p, \mathrm{rep}_{2r}(x)) \le d(p, q) + d(q, x) + d(x, \mathrm{rep}_{2r}(x)) \le r + r + 2r = 2(2r) \ .$$

Thus, our claim holds.

In addition, $D_r \subseteq N_r$, which means that the distance between any two points in $D_r$ is more than $r$. Adding this to the claim, the definition of doubling dimension implies that $D_r$ is of size $2^{O(\mathrm{dim})}$. $\quad\square$

Now, we can discuss the update time for maintaining our data structure after each deletion.

**Lemma A.12.** *Let $n$ be the size of the point set $P_t$. Then after deletion of any point from $P_t$, we can update our data structure in time $2^{O(\mathrm{dim})}\varepsilon^{-2\,\mathrm{dim}}\log\Delta\log n$.*

*Proof.* As we discussed in Section A.2, to maintain the heaps, a $2^{O(\mathrm{dim})}\varepsilon^{-\,\mathrm{dim}}\log n$ factor will be multiplied by the running time of Algorithm 3. Hence, it is enough to show that the running time of Algorithm 3 is $2^{O(\mathrm{dim})}\varepsilon^{-\,\mathrm{dim}}\log\Delta$.

In Algorithm 3, we first call FindNeighbours for $\gamma = 4$ and $\gamma = 1 + (3 + 8\varepsilon)/\varepsilon$ to find sets $S_r$ and $T_r$, which take time $2^{O(\mathrm{dim})}\log\Delta$ and $2^{O(\mathrm{dim})}\varepsilon^{-\,\mathrm{dim}}\log\Delta$, respectively, according to Lemma A.5.

We next iterate over $O(\log\Delta)$ levels and in each level, we iterate over some children of $p$ and the elevated points $E_{r/2}$, and then we call UpdateNeighbourhoodRemove, which takes $O(|B_{p,r}^3| + |B_{p,r}^1|) = O(2^{O(\mathrm{dim})}\varepsilon^{-\,\mathrm{dim}})$. The number of $p$'s children and the elevated points are $2^{O(\mathrm{dim})}$ by Lemma A.2 and Lemma A.10, respectively. Thus, we iterate over $2^{O(\mathrm{dim})}$ points in each level, and in each iteration, all the operations other than UpdateNeighbourhoodElevate can be done in time $2^{O(\mathrm{dim})}$. Besides, the running time of UpdateNeighbourhoodElevate is $O(|T_{r'}|) = O(2^{O(\mathrm{dim})}\varepsilon^{-\,\mathrm{dim}})$. Hence, the total running time of the iteration over the $O(\log\Delta)$ levels is $O(2^{O(\mathrm{dim})}\varepsilon^{-\,\mathrm{dim}}\log\Delta)$. $\quad\square$

**Proof of Lemma 2.3** Now, adding Lemma A.7 and Lemma A.12 together, finishes the proof of Lemma 2.3, which is about the update time of our data structure.

**Lemma 2.3.** *Let $n$ be the size of point set $P_t$. Then after insertion/deletion of any point to $P_t$, we can update our data structure in time $2^{O(\mathrm{dim})}\varepsilon^{-2\,\mathrm{dim}}\log\Delta\log n$.*

## B Omitted Proofs

**Lemma 2.1.** *Let $p \in P_t$ and $r \in R$. Then $d(p, \mathrm{rep}_r(p)) \le 2r$. In particular, $d(p, \mathrm{rep}_{\varepsilon r}(p)) \le 2\varepsilon r$, where $\varepsilon = 2^{\lfloor \log \varepsilon' \rfloor}$ for the error parameter $\varepsilon'$.*

*Proof.* Let $r' \in R$. As $N_{2r'}$ is a $2r'$-net of $N_{r'}$, we know that $d(\mathrm{rep}_{r'}(p), \mathrm{rep}_{2r'}(\mathrm{rep}_{r'}(p))) \le 2r'$. Further, by definition, $\mathrm{rep}_{2r'}(p) = \mathrm{rep}_{2r'}(\mathrm{rep}_{r'}(p))$ and hence, $d(\mathrm{rep}_{r'}(p), \mathrm{rep}_{2r'}(p)) \le 2r'$. Thus,

$$d(p, \mathrm{rep}_r(p)) \le \sum_{\substack{r' < r, \\ r' \in R}} d(\mathrm{rep}_{r'}(p), \mathrm{rep}_{2r'}(p)) \le 2 \sum_{\substack{r' < r, \\ r' \in R}} r' = 2\beta \sum_{\ell < \log \frac{r}{\beta}} 2^\ell \le 2\beta 2^{\log \frac{r}{\beta}} = 2r \ . \quad (1)$$

For $r' = \varepsilon r$, we make a case distinction. If $\varepsilon r \ge \beta$, then $\varepsilon r \in R$ and hence $d(p, \mathrm{rep}_{\varepsilon r}(p)) \le 2\varepsilon r$ by Equation (1). If $\varepsilon r < \beta$, then $d(p, \mathrm{rep}_{\varepsilon r}(p)) = 0 \le 2\varepsilon r$ because $\mathrm{rep}_{\varepsilon r}(p) = p$ by definition. $\quad\square$

**Lemma 2.2.** *Let $r \in R$ and $p \in N_{\varepsilon r}$. The sets $B_{p,r}^1$ and $B_{p,r}^3$ defined above are of size at most $\frac{2^{O(\mathrm{dim})}}{\varepsilon^{\mathrm{dim}}}$. In particular, the size is $O(\varepsilon^{-\,\mathrm{dim}})$ for a constant doubling dimension $\mathrm{dim}$.*

*Proof.* By recursively applying the definition of doubling dimension $i$ times, we can cover a ball of radius $(3 + 8\varepsilon)r$ with at most $2^{i\,\dim}$ balls of radius $\frac{(3+8\varepsilon)r}{2^i}$. For $i = \lceil \log \frac{3+8\varepsilon}{\varepsilon/2} \rceil$ it follows that $B^3_{p,r}$ can be covered by $(2 \cdot \frac{3+8\varepsilon}{\varepsilon/2})^{\dim} \in \frac{2^{O(\dim)}}{\varepsilon^{\dim}}$ balls of radius $\frac{\varepsilon r}{2}$. As $B^3_{p,r} \subseteq N_{\varepsilon r}$, the distance between any pair of points in $B^3_{p,r}$ is at least $\varepsilon r$. Therefore, each of the $\frac{\varepsilon r}{2}$-balls contains at most one point, which finishes the proof. □

**Lemma 3.1.** *Let $k$ be an integer, $0 \leq i \leq k$, and $r \in R$. After the $i$-th iteration of the loop in Line 3, $H_r[y]$ is the total weight of unmarked points in $B^1_{y,r}$ for each $y \in N_{\varepsilon r}$. More formally, $H_r[y] = w_{\varepsilon r}(B^1_{y,r} \cap U_i)$ for any $y \in N_{\varepsilon r}$ after the $i$-th iteration.*

*Proof.* Let $H^i_r$ denote the state of the heap and $M^i$ the set Marked at the end of iteration $i$, and $H^0_r$ and $M^0$ denote the respective sets before entering the for-loop. It can easily be seen that $M^i = \cup_{j \leq i} B^3_{c_j,r}$ for every $i \leq k$ by a simple induction. Initially, $M^0 = \emptyset = \cup_{j \leq 0} B^3_{c_j,r}$. Now assume $M^{i-1} = \cup_{j \leq i-1} B^3_{c_j,r}$ for some $i < k$. Noting that $M^{i-1}$ is only updated in line 7 of MAXCOVERAGE$(k, r)$, Lines 6 and 7 imply that all points from $B^3_{c_i,r}$ that are not in $M^{i-1}$ yet are added to $M^{i-1}$. Therefore, $M^i = M^{i-1} \cup B^3_{c_i,r} = \cup_{j \leq i} B^3_{c_j,r}$. Recall that $U_i = N_{\varepsilon r} \setminus \cup_{j \leq i} B^3_{c_j,r}$. This directly implies that $U_i$ corresponds to the set of points that are still unmarked at the end of iteration $i$, that is $U_i = N_{\varepsilon r} \setminus M^i$. From this, it also immediately follows that $U_i = U_{i-1} \setminus B^3_{c_i,r}$.

Now we can show $H^i_r[x] = w_{\varepsilon r}(B^1_{x,r} \cap U_i)$ for all $i \leq k$. For $i = 0$, the heap is not modified and no points are covered yet, i.e. $H^0_r[x] = w_{\varepsilon r}(B^1_{x,r})$ and $U_0 = N_{\varepsilon r}$. Now assume $H^{i-1}_r[x] = w_{\varepsilon r}(B^1_{x,r} \cap U_{i-1})$ for some $i < k$. Lines 6 to 9 in MAXCOVERAGE$(k, r)$ are equivalent to

$$\forall y \in B^3_{c_i,r} \setminus M^{i-1} \forall x \in B^1_{y,r}: \text{ decrease } H_r[x] \text{ by } w_{\varepsilon r}(y) \ . \tag{2}$$

Neighborhood membership is a symmetric relation in the following sense. If $x \in N_{\varepsilon r}$ is contained within the local neighborhood of $y \in N_{\varepsilon r}$, then the converse is also true: For every $x, y \in N_{\varepsilon r}$ such that $x \in B_{y,r}$ it also holds $y \in B_{x,r}$. Therefore, Equation (2) is equivalent to

$$\forall x \in N_{\varepsilon r} \forall y \in B^3_{c_i,r} \cap B^1_{x,r} \setminus M^{i-1}: \text{ decrease } H_r[x] \text{ by } w_{\varepsilon r}(y) \ ,$$

which means that during iteration $i$, the heap value of $x \in N_{\varepsilon r}$ is reduced by $w_{\varepsilon r}(B^3_{c_i,r} \cap B^1_{x,r} \setminus M^{i-1})$. Because of the equality $B^3_{c_i,r} \cap B^1_{x,r} \setminus M^{i-1} = B^3_{c_i,r} \cap B^1_{x,r} \cap U_{i-1}$, we can deduce

$$H^i_r[x] = H^{i-1}_r[x] - w_{\varepsilon r}(B^3_{c_i,r} \cap B^1_{x,r} \cap U_{i-1}) = w_{\varepsilon r}(B^1_{x,r} \cap U_{i-1}) - w_{\varepsilon r}(B^3_{c_i,r} \cap B^1_{x,r} \cap U_{i-1})$$

$$\overset{(3)}{=} w_{\varepsilon r}(B^1_{x,r} \cap U_{i-1} \setminus (B^3_{c_i,r} \cap B^1_{x,r} \cap U_{i-1})) = w_{\varepsilon r}(B^1_{x,r} \cap U_{i-1} \setminus B^3_{c_i,r})$$

$$= w_{\varepsilon r}(B^1_{x,r} \cap U_i) \ ,$$

where (3) follows from the fact that $w_{\varepsilon r}(X \setminus Y) = w_{\varepsilon r}(X) - w_{\varepsilon r}(Y)$ for $Y \subseteq X \subseteq N_{\varepsilon r}$. □

**Lemma 3.6.** *Let $OPT > 0$ be the optimal radius for the $k$-center clustering of $P_t$ with $z$ outliers, and let $r^*$ be the minimum number in $R$ such that $r^* \geq OPT$. Then FINDCENTERS$(k, z)$ returns a $(3 + 10\varepsilon)\rho$-approximate solution for $k$-center clustering problem with $z$ outliers, where $\rho = \frac{r^*}{OPT}$.*

*Proof.* We first show that $\hat{r} \leq r^*$. To do this, we consider two cases: $\hat{r} > 2$ or $\hat{r} \leq 2$. For the first case, it holds that the *outliersWeight* returned by MAXCOVERAGE$(k, \hat{r}/2)$ is more than $z$, and then we can conduct that $OPT > \hat{r}/2$. It is because otherwise, MAXCOVERAGE$(k, \hat{r}/2)$. *outliersWeight* would be less than $z$ by Lemma 3.5, which is a contradiction. It means that $r^* > \hat{r}/2$, and then according to the definition of $R$ and $r^*$ we have $r^* \geq \hat{r}$. Now we consider the second case that $\hat{r} \leq 2$. Since $OPT > 0$, we have $OPT \geq d_{\min} = 2$. Adding it to $OPT \leq r^*$, which is an assumption of this lemma, we have $\hat{r} \leq 2 \leq OPT \leq r^*$. Thus $\hat{r} \leq r^*$ holds in both cases.

Due to Lemma 3.2 and Lemma 3.5, MAXCOVERAGE$(k, \hat{r})$ returned by FINDCENTERS$(k, z)$ is a set of $k$ centers such that all but at most $z$ points in $P_t$ are within distance $(3 + 10\varepsilon)\hat{r}$ of these centers. Hence, FINDCENTERS$(k, z)$ returns a $\frac{(3+10\varepsilon)\hat{r}}{OPT}$-approximation solution. We also showed that $\hat{r} \leq r^*$, therefore, the approximation ratio is at most $(3 + 10\varepsilon)\rho$, where $\rho = \frac{r^*}{OPT}$. □

**Lemma 3.7.** *Let $OPT > 0$, and let $\varepsilon > 0$ be fixed. We define $m := \lceil 1/\log_2(1+\varepsilon)\rceil$. Suppose we have $m$ parallel instances of our data structure with parameter $\beta = 2^{i/m-1}$ for the $i$-th instance. Then in at least one of the instances, we find a $(3 + O(\varepsilon))$-approximation solution for $k$-center clustering with $z$ outliers by calling $\textsc{FindCenters}(k,z)$.*

*Proof.* Since we assume $OPT > 0$, then we have $OPT \geq d_{\min} = 2$. Let $j^* \in [1, \lceil\log\Delta\rceil]$ be such that $2^{j^*-1} < OPT \leq 2^{j^*}$, and let $i^* \in [1, m]$ be such that $2^{j^*-1+(i^*-1)/m} < OPT \leq 2^{j^*-1+i^*/m}$. For the instance with parameter $\beta = 2^{i^*/m-1}$, we have $\beta \cdot 2^{j^*} = 2^{j^*-1+i^*/m} \in R$. Therefore by Lemma 3.6, $\textsc{FindCenters}(k,z)$ returns a $(3+10\varepsilon)\rho^*$-approximation solution, where $\rho^* = 2^{j^*-1+i^*/m}/OPT$. To finish the proof of Lemma, we show that $\rho^* \leq (1+\varepsilon)$, which immediately implies a $(3 + O(\varepsilon))$-approximation solution. According to the definition of $i^*$ we have $\rho^* = 2^{j^*-1+i^*/m}/OPT \leq 2^{j^*-1+i^*/m}/2^{j^*-1+(i^*-1)/m} = 2^{1/m}$. Besides, $m \geq 1/\log_2(1+\varepsilon)$ holds by the definition for of $m$, which means $1/m \leq \log_2(1+\varepsilon)$. Therefore, $2^{1/m} \leq (1+\varepsilon)$, and then we have $\rho^* \leq 2^{1/m} \leq (1+\varepsilon)$, which finishes the proof. $\square$

**Lemma 3.8.** *Let $u$ be the number of times that $\textsc{MaxCoverage}(k,r)$ updates heap $H$. Then copying heap $H_r$ in Line 2, as well as all these $u$ updates, can be handled in total time $O(u \log n)$.*

*Proof.* Procedure $\textsc{MaxCoverage}(k,r)$ sets $H$ as a copy of $H_r$, and then it decreases the value of some keys in $H$ during the algorithm. Note that it never inserts or deletes a new key to $H$, and the only possible update is decreasing the value of a key. We claim that it is not necessary to really copy the heap $H_r$, which takes $O(|H_r|) = O(n)$ time. Instead, we apply all the updates on $H_r$ and then undo them at the end of $\textsc{MaxCoverage}(k,r)$. Although the procedure can change $H_r$ as we defined, we guarantee that $H_r$ is the same at the start and end of the procedure. To do this, we keep a record of all $u$ updates, and at end of $\textsc{MaxCoverage}(k,r)$, we undo all these $u$ updates, which means increasing the value of each updated key with the decreased value. Since the number of updates is $u$, then we can do all the reverse updates in $O(u \log |H_r|) = O(u \log n)$. $\square$

**Lemma 3.9.** *Let $k$ and $z$ be two given parameters. Then $\textsc{FindCenters}(k,z)$ described in Algorithm 1 has a time complexity of $2^{O(\dim)}\varepsilon^{-2\dim}k \log n \log\log\Delta$.*

*Proof.* $\textsc{FindCenters}(k,z)$ perform a binary-search on $R$ and in each search, it invokes $\textsc{MaxCoverage}$ two times. Therefore, $\textsc{MaxCoverage}$ is called $O(\log|R|)$ times in total, which is $O(\log\log\Delta)$ as $|R| = O(\log\Delta)$. Hence, it remains to show that the running time of $\textsc{MaxCoverage}(k,r)$ is $2^{O(\dim)}\varepsilon^{-2\dim}k \log n$.

Now, we discuss the running time of $\textsc{MaxCoverage}(k,r)$. This procedure consists of $k$ iterations, in which we iterate on all elements $y \in B^3_{c_i,r}$. Recall that Lemma 2.2 states $B^3_{c_i,r}$ is of size at most $\frac{2^{O(\dim)}}{\varepsilon^{\dim}}$. Then for any $y$, we iterate on all elements $x \in B^1_{y,r}$ and update $H[x]$ in Line 9. Again by Lemma 2.2, the size of $B^1_{y,r}$ is at most $\frac{2^{O(\dim)}}{\varepsilon^{\dim}}$. Therefore, we update heap $H$ at most $\left(\frac{2^{O(\dim)}}{\varepsilon^{\dim}}\right)^2 \cdot k$ times. Then by Lemma 3.8, these heap updates and also copying heap $H_r$ in Line 2 are done in total time of $2^{O(\dim)}\varepsilon^{-2\dim}k \log n$. $\square$

## Proof of Theorem 1.1.

**Theorem 1.1.** *Let $(M, d)$ be a metric space of bounded doubling dimension $dim$, and let $\varepsilon > 0$ be an error parameter. There exists a deterministic dynamic algorithm that allows the insertion or deletion of points from $M$ using worst-case $\varepsilon^{-O(\dim)} \log n \log\Delta$ update time. Moreover, at any time $t$, it can be queried by parameters $k$ and $z$ to compute a $(3+\varepsilon)$-approximate solution for the $k$-center problem with $z$ outliers of $P_t$ using worst-case $\varepsilon^{-O(\dim)}k \log n \log\log\Delta$ query time, where $P_t$ is the set of points that are inserted but not deleted up to time $t$, and $n$ is the size of $P_t$.*

*Proof.* We maintain $m = \lceil 1/\log_2(1+\varepsilon)\rceil$ instances of our data structure with parameter $\beta = 2^{i/m-1}$ for the $i$-th instance. We first discuss the approximation ratio. If $OPT > 0$, then we find a $(3 + O(\varepsilon))$-approximation solution by Lemma 3.7. Otherwise if $OPT = 0$, then by Lemma 3.5, the *outliersWeight* returned by $\textsc{MaxCoverage}(k,0)$ is at most $z$. In this case, we return a solution with radius 0 in Line 2, which is an exact solution.

Now we discuss the update time. If we maintain $m = O(1/\log(1+\varepsilon))$ instances of our data structure in parallel, it incurs an $O(1/\log(1+\varepsilon)) = O(\varepsilon^{-1})$ factor in the time complexity. In each instance, the update time per insert/delete is $2^{O(\dim)}\varepsilon^{-2\dim}\log\Delta\log n$ by Lemma 2.3, and the $k$ centers can be computed in time $O(2^{O(\dim)}\varepsilon^{-2\dim}k\log n\log\log\Delta)$ by Lemma 3.9 Therefore, we handle each insert/delete in $2^{O(\dim)}\varepsilon^{-(2\dim+1)}\log\Delta\log n = \varepsilon^{-O(\dim)}\log n\log\Delta$ total time and we can find the centers in $O(2^{O(\dim)}\varepsilon^{-(2\dim)+1}k\log n\log\log\Delta) = \varepsilon^{-O(\dim)}k\log n\log\log\Delta$ total time. $\quad\square$

