# OpenReview forum: "Faster Query Times for Fully Dynamic $k$-Center Clustering with Outliers"
_NeurIPS.cc/2023/Conference — NeurIPS 2023 poster_

### Official Review · Reviewer_o6fX · 2023-07-02

**Soundness:** 4 excellent
**Presentation:** 4 excellent
**Contribution:** 3 good
**Rating:** 6
**Confidence:** 4

**Summary:**

This paper studies dynamic algorithms for k-center with outliers in a general metric space with bounded doubling dimension, denoted as dim. In the k-center with z outliers problem (which they call (k, z)-center problem), the goal is to find a set of k points C, such that the maximum distance from the data points to the center set is minimized, where the maximization is taken only over the n - z closest data points to C. In terms of the dynamic model, the paper assumes oracle access to the distance function, subject to point insertion and deletions. The query is to return a list of approximate centers, which is of size (at most) k.

The main result is a dynamic algorithm that can handle each update in time eps^{-dim} log n, which is independent of k, and can report a (3 + eps)-approx solution in eps^{-dim} k log n time. The running time is completely independent of z, and for the query in linear in k which improves the (k + z)^2 query time from previous works significantly.

The techniques are based on previous navigating net/cover tree structures. However, nontrivial steps are employed to obtain the improved bounds.

**Strengths:**

- The result is strong. In particular, the running times are independent of z, and the query time is only linear in k which improves over (k + z)^2 as in previous works. I find this improvement significant.
- The comprehensive comparison to various recent works helps a lot in gauging the novelty of this work.
- Technically, although the approach is not novel in terms of overall framework, to fit ideas from several different works together is a nontrivial task, and I find it interesting.

**Weaknesses:**

- The setting of doubling metrics is only partially motivated. Indeed, Bateni et al. [1] showed \Omega(k) is necessary for handling updates in general metrics, but what about the dependence in z? It seems removing the dependence in z in general metrics is still an important goal of study?
- There is no empirical evaluation of your algorithm, which may be important for some in the NeurIPS community.
- I don't find a statement of the main theorem. A rigorous statement may be helpful for people who would like to cite/use your paper without going too much into detail. Also, this makes it somewhat difficult to read your proofs, since it's easy to get lost in the series of lemmas and I don't see the big picture.
- The presentation is not very friendly for non-experts. People without sufficient background in doubling metrics and k-center clustering may find it difficult to understand the paper. I personally only manage to check part of the proofs, although I think it overall makes sense.


**Questions:**

- Table 1, line 2, eps^d -> eps^{-d}. Also line 1 is using 1 / eps, which is inconsistent with eps^{-d} style.
-  Page 2, line 57: I agree that wlog you can assume d_\min = 2 if it is the static setting, where P is given. However, in the dynamic setting, the aspect ratio and/or d_min could change very significantly. Do you assume the d_min is always = 2, and that the aspect ratio is always bounded, at any time step?
- Did you define doubling dimension at all? Also, in line 175, you mentioned it has 2^{O(dim)} children. This is probably standard, but it still requires some explanation, or at least requires a reference.
- In sec 2 you defined a data structure. It would be helpful to state what updates/queries it supports and what's the guarantee in e.g. a lemma statement.
- The "Overview" of Sec 3 seems too detailed and without a big picture. For example, it starts with a discussion of what we do for the heaps, but did not explain what's the overall purpose of this MaxCoeverage procedure.

**Limitations:**

I don't find a clear discussion these. But I suppose there is no potential negative societal impact.

---

> ### Author Rebuttal · Authors · 2023-08-09
>
> **Comment:** I don't find a statement of the main theorem. A rigorous statement may be helpful for people who would like to cite/use your paper without going too much into detail. Also, this makes it somewhat difficult to read your proofs, since it's easy to get lost in the series of lemmas and I don't see the big picture.
>
> **Response:** The main theorem is formally stated on pages 2-3 (Theorem 1.1). The proof of this theorem can be found in the appendix (page 18 of the full version).
>
>
> **Question**: Page 2, line 57: I agree that wlog you can assume $d_{\min} = 2$ if it is the static setting, where P is given. However, in the dynamic setting, the aspect ratio and/or $d_{\min}$ could change very significantly. Do you assume the $d_{\min}$ is always = 2, and that the aspect ratio is always bounded, at any time step?
>
> **Response:** To have a dynamic aspect ratio, we can assume that the navigating net has an infinite number of levels (by defining $R:=\set{0} \cup \set{\beta \cdot 2^{\ell}| \ell \in \mathbb{Z}}$), however, only $O(\log{\Delta})$ levels are active at a time. We consider the highest level with $n$ nodes for $d_{\min}$ and the lowest level with only one node for $d_{\max}$. Thank you for bringing this to our attention. We will clarify it in the revised version.
>
> **Question**: Did you define doubling dimension at all? Also, in line 175, you mentioned it has $2^{O(\text{dim})}$ children. This is probably standard, but it still requires some explanation, or at least requires a reference.
>
> **Response:** Doubling dimension is defined in Line 54. We will highlight it with more details in the revised version.
>
> **Question**: In sec 2 you defined a data structure. It would be helpful to state what updates/queries it supports and what's the guarantee in e.g. a lemma statement.
>
> **Response:** The update time of our data structure is specified in Lemma 2.3. We will explain the update/query methods that our data structure supports in greater detail in the revised version.
>
> **Question**: The "Overview" of Sec 3 seems too detailed and without a big picture. For example, it starts with a discussion of what we do for the heaps, but did not explain what's the overall purpose of this Max Coverage procedure.
>
> **Response:** Thank you for bringing this to our attention. We will provide a big picture of our algorithm in the revised version.

---

> > ### Comment · Reviewer_o6fX · 2023-08-18
> >
> > Thanks for the response, and most comments are addressed (or promised to be addressed in the next revision).
> >
> > But you still missed this important reference though:
> >
> > Bounded Geometries, Fractals, and Low-Distortion Embeddings. Anupam Gupta, Robert Krauthgamer and James R. Lee. FOCS 2003.
> >
> > This paper is the standard reference for doubling metrics and packing property etc. Please cite.
> >
> > Another small issue is that the definition of the aspect ratio is still not clear: are you assuming the dataset always has a bounded aspect ratio \Delta over all updates (which is known in advance), or \Delta is the aspect ratio of the current dataset dynamically?

---

> > > ### Author Response · Authors · 2023-08-20
> > >
> > > Thank you for pointing out this reference. We will add it to the final version.
> > >
> > > Like the other works [1, 3, 4, 7, 10, 17], we assumed that the aspect ratio $\Delta$ is bounded over all updates and is known in advance. However, we believe that it is also possible that our bounds also hold for a dynamic aspect ratio $\Delta$. Indeed, let $\Delta_t$ and $\Delta_{t-1}$ be the spread ratios after the update at time $t$ and $(t-1)$, respectively. Then, for the update and query at time $t$, we can update the current bounds by replacing $\Delta$ with ($\Delta_t + \Delta_{t-1}$).

---

### Official Review · Reviewer_RPz3 · 2023-07-05

**Soundness:** 3 good
**Presentation:** 2 fair
**Contribution:** 3 good
**Rating:** 6
**Confidence:** 3

**Summary:**

The main contribution of this paper is the proposal of a new approximate algorithm for "fully dynamic k center with outliers" problem in metric space with low doubling dimension. The approximation ratio is 3+$\epsilon$. The meaning of "fully dynamic" is that the algorithm should be able to efficiently adjust its output to accommodate new data when deletion or insertion operations are performed on the dataset. To achieve this goal, we need to maintain a data structure to handle the addition and deletion of data, and to efficiently provide an approximate solution for the k center with outliers problem at any time. According to this paper,  the time complexity of query (given the data structure has been established, to generate a solution for the k center with outliers problem) is $\epsilon ^{-O(\mathtt{dim})}k \log n \log \log \Delta$, where $\Delta$ is the ratio between the maximum and minimum distance between two points in  dataset, $\mathtt{dim}$ is the doubling dimension of the metric space. And the time complexity of insertion/deletion  is $\epsilon ^{-O(\mathtt{dim})}\log n \log \Delta$.

**Strengths:**

1. Faster binary search for a specific problem. The "Query" algorithm in the paper essentially uses binary search to find the appropriate clustering radius, for example, by searching within the interval $[0, \Delta]$. This step usually takes $O(\log \Delta)$ time, but the technique used in this paper can reduce this complexity to $O(\log \log \Delta)$. This technique appears to be applicable for solving some problems other than the k center with outliers problem.
2. Lower time complexity in some cases.  The time complexity of addition/deletion operation is independent of the number of cluster centers k and the number of outliers z in the data. And the time complexity of querying is linearly related to the k. This fact gives the algorithm proposed in this paper a significant advantage in terms of running time when k or z is very large.

3. State-of-the-art approximation ratio. The approximation ratio of this algorithm achieves the current state-of-the-art results ($3+\epsilon$).

**Weaknesses:**

1. Time complexity. Table 2 in this paper presents a comparison of the time complexity of the proposed algorithm with some existing algorithms for solving the dynamic k center with outliers problem. The table reveals that the previous works (such as [3, 16, 7] in the references) have update and query time complexities that are independent of n, whereas the proposed algorithm has a linear dependence on log n for both update and query. As a result, the proposed algorithm may have slower running times when n is large and k and z are relatively small.
2. Complex data structures. The algorithm in the text uses relatively complex data structures, which may not be conducive to the implementation of the algorithm and the final actual running time. I am wondering about the space complexity.

**Questions:**

1. Line 148, "We utilize $\beta$ to define $R:=\{0\}\cup \{\beta \cdot 2^l | 0 \le l \le \lceil \log _2 \Delta \rceil \}$ as the set of levels.".  Does '$l$' have any restrictions here? I guess '$l$' should only take integers? Otherwise, if '$l$' is a real number in this range, the time complexity of Algorithm 1 will be linearly related to $\log \Delta$.

2. You mentioned in the paper that the time complexity of Algorithm 1 is linear with respect to $\log \log \Delta$, which is an interesting conclusion because traditional binary search results in time complexity linearly related to log Delta. I noticed that you used some techniques to make this conclusion hold. I wonder if these techniques can be applied to similar binary search scenarios? For example, in [3], the k-center with outliers algorithm based on binary search, can we also obtain a time complexity linearly related to $\log \log \Delta$?

**Limitations:**

The work is mostly theoretical, and SODA/ICALP/ESA may be better venue for this paper?

---

> ### Author Rebuttal · Authors · 2023-08-09
>
> **Comment:** Time complexity. Table 2 in this paper presents a comparison of the time complexity of the proposed algorithm with some existing algorithms for solving the dynamic $k$ center with outliers problem. The table reveals that the previous works (such as [3, 16, 7] in the references) have update and query time complexities that are independent of $n$, whereas the proposed algorithm has a linear dependence on $\log{n}$ for both update and query. As a result, the proposed algorithm may have slower running times when $n$ is large and $k$ and $z$ are relatively small.
>
> **Response:** Thank you for pointing this out. Please note that for $n$ distinct points, the spread ratio satisfies $\Delta \geq n^{1/\text{dim}}$ and all mentioned algorithms involve a dependency on $\log{\Delta} \geq \log{n} / \text{dim}$. For constant $\dim$, our query time can be bounded by $ O(\epsilon^{-\text{dim}} k \log{\Delta} \log\log{\Delta})$, which favorably compares to [16] for example in the regime of $z = O(\epsilon n)$ where our query time still has a logarithmic dependency on $n$ while [16] would have a quadratic dependency on $n$. Similarly, if either $k$ or $z$ is mildly dependent on $n$, our query time is significantly better than the query time in [16]. Indeed, already for $k \in O(\log{n})$ and $z\in \Omega(\sqrt{n})$, the query complexity of [16] would be linear in $n$ compared to the poly-logarithmic dependency in terms of $n$ for our query time.
>
> In addition, please note that [7] is only for the Euclidean case and is randomized while our algorithm works for doubling metrics and is deterministic, which implies that it can be used for adaptive adversaries rather than only for oblivious adversaries. Also, the update time of [7] is $O((k/\epsilon^\text{ddim}+z)\log^4{(\frac{k\Delta}{\epsilon\delta})})$ which is significantly worse than the update time or the query time of our algorithm. Moreover, [3] provides a $14$-approximation instead of a $3$-approximation. As mentioned, both algorithms also include a term that depends on n.
>
>
> **Question:** Line 148, "We utilize $\beta$ to define $R := \\{0\\} \cup \\{\beta\cdot 2^\ell | 0 \leq \ell \leq \log\Delta \\}$ as the set of levels.". Does '$\ell$' have any restrictions here? I guess '$\ell$' should only take integers? Otherwise, if '$\ell$' is a real number in this range, the time complexity of Algorithm 1 will be linearly related  to $\log \Delta$.
>
> **Response:** Yes, you are right, $\ell$ can take only integers.  Thank you for bringing this to our attention. We will clarify it in the revised version.
>
>
>
>
>
> **Question:** You mentioned in the paper that the time complexity of Algorithm 1 is linear with respect to $\log{\log{\Delta}}$, which is an interesting conclusion because traditional binary search results in time complexity linearly related to log Delta. I noticed that you used some techniques to make this conclusion hold. I wonder if these techniques can be applied to similar binary search scenarios? For example, in [3], the k-center with outliers algorithm based on binary search, can we also obtain a time complexity linearly related to $\log{\log{\Delta}}$?
>
> **Response:** This is because the set that we are searching over only has size $\log \Delta$. That is because the radii that we are checking are approximate, so there are only $O(\log \Delta)$ candidates of the form $(1+\epsilon)^i$ to be checked to cover the interval $[1,\Delta]$. For classical $k$-center, when we are considering a set of $\Theta(n^2)$ distance candidates (all pairwise distances), then doing binary search over the sorted list of candidates only leads to a $\log n$ and that cannot be improved to $\log \log n$ because it is a different candidate set. We had a look into [3] after your comment and indeed believe that for the query time of their algorithm, $\log \Delta$ can be improved to $\log \log \Delta$ in the same fashion.

---

> > ### Comment · Reviewer_RPz3 · 2023-08-19
> >
> > Thanks for your response, and I am willing to increase my score to 6.

---

### Official Review · Reviewer_JTLo · 2023-07-06

**Soundness:** 3 good
**Presentation:** 3 good
**Contribution:** 3 good
**Rating:** 7
**Confidence:** 3

**Summary:**

This paper presents a new algorithm for fully dynamic $k$-center clustering with outliers in spaces with bounded doubling dimension $dim$ and aspect ratio $\Delta$, where the number of outliers $z$, can be specified at query time. While the approximation ratio of $3+\epsilon$ is on par with previous results, it trades a $\log(n)$ dependencies for a removal of a $k$ factor from update and query time and moving $z$ from the problem parameterization to the query specification. In particular, the update time is $1/\epsilon^{O(dim)} \log n \log \Delta$, and the query time is $k/\epsilon^{O(dim)} \log n \log \log \Delta$. Previous algorithms differ in the query time by at least a factor of $(k+z^2) / log(n)$.

The algorithm builds on the previous result by Pellizzoni et al. and improves its internal tree-based data structure by augmenting additional information. The tree imposes a hierarchy on the input points and stores, for each representative point of a node, a small and a large neighborhood and a list of child points ordered by the number of their descendants. By maintaining the hierarchy for $\log \Delta$ choices of the neighbhorhoods' radius (and the fact that the radius of the neighbhorhood scales linearly with the $k$-center cost), the ordered lists allow to construct a $k$-center solution in-place at node level.

Overall, I think this paper showcases nicely how one self-contained improvement to an existing algorithm can lead to an interesting property like time complexity independet of $z$ and actually moving the specification of $z$ from problem paramters to query arguments. The resulting bounds are nice and are an improvement in the interesting case that $k \in \Omega(\log n)$ or $z$ is a variable.

**Strengths:**

* Update and/or query time improvement over previous results for $k \in \Omega(\log n)$.
* The algorithm solves the dynamic $k$-center problem with $z$ outliers for all $z$ simulteanously.
* The time complexities are independent of $z$.

**Weaknesses:**

* The time complexities are not better for all settings, and the update time on its own is always slightly worse compared to one other existing algorithm.

**Questions:**

-

**Limitations:**

-

---

> ### Author Rebuttal · Authors · 2023-08-09
>
> We greatly appreciate your time in reviewing our paper and your feedback.

---

### Official Review · Reviewer_bgty · 2023-07-10

**Soundness:** 4 excellent
**Presentation:** 3 good
**Contribution:** 4 excellent
**Rating:** 6
**Confidence:** 5

**Summary:**

In k-center with z outliers problem, the input is a data set X that lies in metric space and the goal is to find a center set C of size k such that the z+1 largest distance from points in X to C is minimized. This paper stuided a dynamic version of this paper where data point can be inserted and deleted but we should output an approximate solution to the problem at any time.

The main result of this paper is $3+\epsilon$ approximation dynamic algorithm that supports an $\epsilon^{-O(\mathrm{ddim})}\log n \log \Delta$ update time and an $\epsilon^{-O(\mathrm{ddim})}k\log n \log \log \Delta$ where $\mathrm{ddim}$ is the doubling dimension of metric space, $n=|X|$, and $\Delta$ is the ratio between the maximum and minimum pariwise distance of $X$.

This result does not significantly improve previous results. For example, [16] already gave a dynamic algorithm with an $\epsilon^{-O(\mathrm{ddim})}\log \Delta$ update time, an $\epsilon^{-O(\mathrm{ddim})}(k+z)^2\log \Delta$ query time, and the same approximation ratio. The new algorithm only trade a $poly(k+z)$ factor with an $\log n$ in the query time. Authors claimed the work is finished indepdently from [16] while even comparing with a weaker and ealier result [7], the new bounds do not seem to be very strong.

Moreover, I do not see the main technical contributions of this paper after reading the first few pages. Authors introduce a few existing techniques of other papers but skip to technical parts directly. I suggest to add a section named "Technical Contribution" to feed the readers with spoons.

**Strengths:**

1. A new algorithm for robust dynamic k-center in doubling metric space.

**Weaknesses:**

1. The algorithm only achieves similar guarantee, comparing with existing results.


**Questions:**

1. The algorithm seems to build on a few existing techniques and the main technical contributions of this paper are not very clear to me. Can you summary your main technical contributions?

**Limitations:**

I do not see potential negative societal impact.

---

> ### Author Rebuttal · Authors · 2023-08-09
>
>
> **Comment:** This result does not significantly improve previous results. For example, [16] already gave a dynamic algorithm with an $\epsilon^{-O(\text{ddim})} \log{\Delta}$ update time, an $\epsilon^{-O(\text{ddim})} (k+z)^2 \log{\Delta}$ query time, and the same approximation ratio. The new algorithm only trade a $poly(k+z)$ factor with an $\log{n}$ in the query time. Authors claimed the work is finished independently from [16] while even comparing with a weaker and earlier result [7], the new bounds do not seem to be very strong.
>
> **Response:** Interestingly, if we assume that the $n$ input points are distinct, then $\Delta \geq n^{1/\text{ddim}}$, so $\log{n} \leq \text{ddim}\cdot\log{\Delta}$. Thus, our query time can also be bounded by $ O(\epsilon^{-\text{ddim}} k \log{\Delta} \log\log{\Delta})$ for constant $\text{ddim}$ while the running time of [16] is $O(\epsilon^{-\text{ddim}}(k+z)^2 \log{\Delta})$. Thus, we are replacing a $(k+z)^2$ by a $k\cdot\log\log{\Delta}$ factor. Now, we consider the following regimes:
> 1. $k$ and $z$ are constants: For this case, our query time and the query time in [16] are comparable, [16] is slightly better by a $\log
> \log \Delta$ factor.
> 2. If either $k$ or $z$ is mildly dependent on $n$, our query time is significantly better than the query time in [16]. In fact, already for $k \in O(\log{n})$ and $z\in \Omega(\sqrt{n})$, the query complexity of [16] would be linear in $n$ compared to the poly-logarithmic dependency in terms of $n$ for our query time.
> 3. More interestingly, for a realistic regime where $z$ is an $\epsilon$-fraction of $n$, the query time of the dynamic algorithm presented in [16] has quadratic dependency ($n^2$) in terms of $n$ while our query time still has logarithmic dependency in terms of $n$ and linear dependency in terms of $k$.
>
> About the comparison with [7], we would like to mention a few differences between our work and [7]:
> - First, their algorithm is only applicable to the Euclidean metric space, however, our algorithm can deal with a more general case of metric spaces with bounded doubling dimension.
> - Second, the algorithm in [7] is randomized and only works against oblivious adversaries while our algorithm is deterministic and works against adaptive adversaries as well as oblivious ones.
> - Third, the update time of [7] is $O((k/\epsilon^\text{ddim}+z)\log^4{(\frac{k\Delta}{\epsilon\delta})})$ (as stated in Table 2) which is significantly worse than the update time or the query time of our algorithm.
>
>
> **Question:** The algorithm seems to build on a few existing techniques and the main technical contributions of this paper are not very clear to me. Can you summarize your main technical contributions?
>
> **Response:** We devise a novel approach to make the greedy algorithm for $k$-center with $z$ outliers **dynamic** while the previous dynamic algorithms extract a coreset and run the greedy algorithm on this coreset. We next explain this difference in detail.
> - **Previous dynamic algorithms:** Known dynamic algorithms [3,7,16] for the $k$-center problem with $z$ outliers maintain a coreset after every update. In particular, in [16], they extract the coreset by simply reading the solution from the cover tree. To extract an approximate solution for this problem, one needs to run a known (offline) greedy algorithm on this coreset. In this way, the query complexities of those algorithms are dominated by the running time of the greedy algorithm.
> - **Our dynamic algorithm:** The novelty of our dynamic algorithm is that we make the greedy algorithm itself dynamic. To this end, we use heap data structures to compute a ball that covers the maximum number of points, and dynamic neighborhood sets to obtain points in the expanded maximum ball and update their corresponding keys in the heap to recursively find the next maximum balls.

---

> > ### Comment · Reviewer_bgty · 2023-08-12
> >
> > I have read the feedback as well as other reviews. My initial concern regarding technical contributions have been resolved and I decide to change my evaluation.

---

### Official Review · Reviewer_k1og · 2023-07-27

**Soundness:** 3 good
**Presentation:** 4 excellent
**Contribution:** 3 good
**Rating:** 7
**Confidence:** 3

**Summary:**

This paper consider the $k$-center problem with outliers in the fully dynamic setting (insertions and deletions of points).

The assumption is that the metric space has bounded doubling dimension (as a side note: some assumption is necessary to get better bounds due to known lower bounds).

The main result of the paper is to give an algorithm with the following guarantees:

- Worst-case  update time  exp(double-dimension) log n log (Delta)
- Worst-case query time: exp(double-dimension)  k log n log log Delta

Note that the query time is almost optimal in the sense that $k$ is needed to return $k$ centers.  Independent concurrent work has achieved similar guarantees. In particular, [16] uses similar ideas but pay an k^2 in query time.
The idea of this paper and [16] is to use "Navigating sets" as previously used. Now an issue is that they don't work off the shelves for the outlier settings. [16] solves this by showing that a certain level forms a coarset and then runs the 3-approximation algorithm Charikar et al. To run this algorithm from scratch incurs a running time overhead. Instead the current paper includes smart information in their datastructrue that avoids running this algorithm from scratch leading to the better running time.

**Strengths:**

- An improvement on a natural problem k-center with outliers in the dynamic setting.

- Close to optimal query time (is the exponential dependency on doubling dimension necessary?)

- Smart algorithmic techniques.

**Weaknesses:**

- Quite a complex algorithm in the end.

- The improvement is quite specialized: query time for the case of double dimensions. Yes, I know that some restrictions on the input is necessary so cannot complain too much but I do think this decreases the general appeal of the paper.

- Would 2-approximation be possible? This is interesting as the best "combinatorial" approximation algorithm for k-center with outliers is 3...



**Questions:**

- What factors in your running times are necessary? Is the exponential dependency on doubling dimension necessary etc?

Minor things:

- line 37 "which gives"

- Thm 1.1. shouldn't it be insertions and deletions instead of insertions or deletions?





**Limitations:**

-

---

> ### Author Rebuttal · Authors · 2023-08-09
>
> **Question:** What factors in your running times are necessary? Is the exponential dependency on doubling dimension necessary etc?
>
> **Response:** These are interesting questions. The exponential dependency on the doubling dimension seems unavoidable. We will pose this question as an open problem for future work. In general, it would be interesting to devise a lower bound for the query time complexity of dynamic algorithms for this problem.
>
> We should mention that there is a lower bound on the storage needed for obtaining $(1+\epsilon)$-approximations in the streaming model via coresets proven by De Berg et al [7] (below). In particular, they show that any dynamic deterministic algorithm for this problem needs storage of $\Omega( (k/\epsilon^{\text{dim}})\log{\Delta} + z )$. That is, an exponential dependency on doubling dimension and a logarithmic dependency on the aspect ratio ($\Delta$) in the storage of any streaming algorithm is necessary. However, the streaming setting is incomparable to the dynamic model and it is not clear if a space requirement of a streaming algorithm can be transferred to the query time requirement of a dynamic algorithm. In fact, the streaming algorithm needs $\Omega(z)$ bound in the space, but we do not have such a term in the query time of our dynamic algorithm.
>
> **Reference:**
> [7] Mark de Berg, Leyla Biabani, Morteza Monemizadeh: k-Center Clustering with Outliers in the MPC and Streaming Model. IPDPS 2023

---

> > ### Comment · Reviewer_k1og · 2023-08-16
> >
> > Interesting, thanks!

---

### Author Rebuttal · Authors · 2023-08-09

We would like to thank the reviewers. We appreciate your time in reviewing our paper and your feedback. Here are our responses to the questions. We will address any minor errors like typos and proposed references promptly.

---

### Decision · Program_Chairs · 2023-09-21

**Decision:**

Accept (poster)

**Comment:**

All reviewers support the paper due to the importance of the problem as well as the significant improvement in the runtime compared with previous works, which is now nearly optimal in certain parameters.